# Enrichment and sensing tumor cells by embedded immunomodulatory DNA hydrogel to inhibit postoperative tumor recurrence

Danyu Wang [1], Jingwen Liu[1], Jie Duan[1], Hua Yi[1], Junjie Liu [1,2,3], Haiwei Song [1,4] ✉, Zhenzhong Zhang[1,2,3,5] ✉, Jinjin Shi [1,2,3] ✉ & Kaixiang Zhang [1,2,3] ✉

Postoperative tumor recurrence and metastases often lead to cancer treatment failure. Here, we develop a local embedded photodynamic immunomodulatory DNA hydrogel for early warning and inhibition of postoperative tumor recurrence. The DNA hydrogel contains PDL1 aptamers that capture and enrich in situ relapsed tumor cells, increasing local ATP concentration to provide a timely warning signal. When a positive signal is detected, local laser irradiation is performed to trigger photodynamic therapy to kill captured tumor cells and release tumor-associated antigens (TAA). In addition, reactive oxygen species break DNA strands in the hydrogel to release encoded PDL1 aptamer and CpG, which together with TAA promote sufficient systemic antitumor immunotherapy. In a murine model where tumor cells are injected at the surgical site to mimic tumor recurrence, we find that the hydrogel system enables timely detection of tumor recurrence by enriching relapsed tumor cells to increase local ATP concentrations. As a result, a significant inhibitory effect of approximately 88.1% on recurrent tumors and effectively suppressing metastasis, offering a promising avenue for timely and effective treatment of postoperative tumor recurrence.

Surgical resection is the first choice for treating solid tumors, yet postoperative tumor recurrence and metastasis often lead to treatment failure[1-3]. Tumor recurrence is difficult to detect with conventional tumor diagnostic methods, leading to a large proportion of patients developing metastases before validated warning[4,5]. Furthermore, the current post-operative cancer treatments, such as systemic chemotherapy and radiotherapy, can control metastases to some extent, but they have low remission rates and collateral damage to normal tissues[6-8]. Considering the weakness of post-operative patients, systemic chemotherapy and radiotherapy are usually administered after a period of recovery from surgery, which often misses the best time to destroy residual tumor cells[9-11]. Therefore, more sensitive detection of tumor recurrence and timely prevention of postoperative secondary metastasis are urgently needed.

Currently, the clinical method for the detection of malignant tumor recurrence and metastasis mainly relies on imaging

[1]School of Pharmaceutical Sciences, Zhengzhou University, Zhengzhou 450001, China. [2]Key Laboratory of Targeting Therapy and Diagnosis for Critical Diseases, Zhengzhou 450001, China. [3]Key Laboratory of Advanced Drug Preparation Technologies, Ministry of Education, Zhengzhou 450001, China. [4]Institute of Molecular and Cell Biology, Agency for Science, Technology, and Research (A*STAR), Singapore 138673, Singapore. [5]State Key Laboratory of Esophageal Cancer Prevention & Treatment, Zhengzhou 450001, China. ✉e-mail: haiwei@imcb.a-star.edu.sg; zhangzhenzhong@zzu.edu.cn; shijinyxy@zzu.edu.cn; zhangkx@zzu.edu.cn

examinations and tumor biomarker monitoring[12–14]. Imaging tests sometimes show false positive signals to post-operative inflammation, while endoscopic examination carries the risk of infection[15–17]. On the other hand, due to the small number of recurrent tumor cells and the low concentration of tumor-specific biomarkers, blood detection of tumor recurrence is challenging[12]. To address this issue, we propose a hypothesis that in situ enrichment of tumor cells triggers local warning signals that may facilitate early recurrence detection. Moreover, residual tumor cells might remain at the surgical margins or enter the circulation after surgery, raising the risk of cancer recurrence and metastasis[18–20]. Capturing and killing tumor cells in situ also has great practical significance for controlling tumor metastasis.

In addition to tumor detection, postoperative consolidation therapy for cancer patients is also important[5]. The poor immunogenicity and immunosuppression at the site of tumor resection further contribute to the development of the tumor, and the best time for treatment is often missed when it reaches the middle or late stages[21,22]. With the rapid progression of cancer immunotherapy, embedded hydrogels with local immunomodulatory ability to boost systemic antitumor immune responses have attracted much attention[23–27]. Studies have shown that immunotherapeutic hydrogels can arouse the host's innate and adaptive immunity to inhibit both local tumor recurrence and possible metastasis, showing exciting results in treating a range of cancer types, such as lung cancer[28,29], breast cancer[30,31]

and melanoma[32,33]. Besides, given the complex mechanism of post-surgical tumor metastasis, it is difficult to achieve an effective treatment with a single mechanism of intervention[34]. Therefore, the combination of synergistic immunotherapies such as phototherapy and immunotherapy, which have high spatiotemporal selectivity and low side effects, is an attractive approach for cancer treatment.

In this work, we construct a diagnostic and therapeutic integrated postoperative embedded DNA hydrogel for tumor recurrence warning and spatiotemporally controlled photodynamic immunotherapy (Fig. 1). The DNA hydrogel contains PDL1 aptamers that can capture and enrich early relapsed tumor cells[35–37]. The aggregated tumor cells significantly improve local ATP concentration, and a circular ATP sensor is designed to anchor on DNA hydrogel to provide a timely warning signal. In order to provide timely and effective treatment upon signal detection, we also load photosensitizer Ce6 and immune adjuvant CpG in the hydrogel[38–40]. When a positive signal is detected, local laser irradiation was performed to trigger PDT and generate ROS to damage the captured tumor cells and release tumor-associated antigens[41,42]. Meanwhile, ROS induces hydrogel self-disassembly, releasing encoded PDL1 aptamer and CpG, exerting immune checkpoint blockade and immune activation to induce immune factor secretion and adequate systemic antitumor immunotherapy[43]. In summary, the developed DNA hydrogel enables timely monitoring of tumor recurrence by enriching tumor cells and can activate powerful

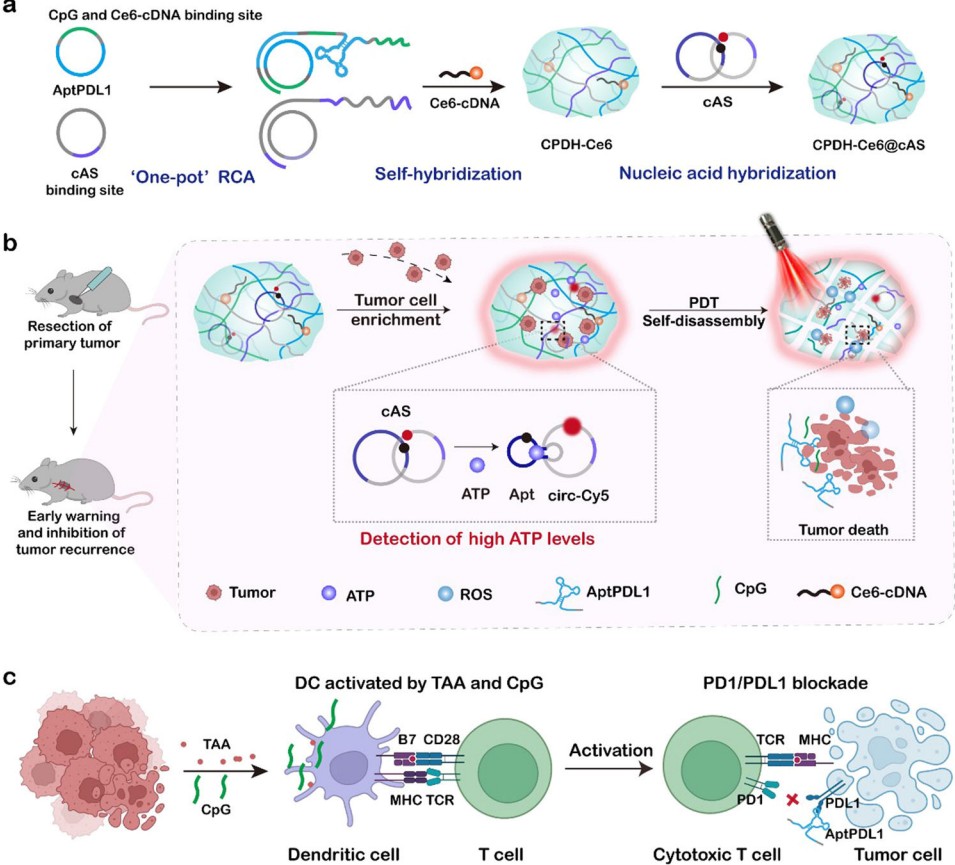

**Fig. 1 | Embedded photoresponsive immunotherapy hydrogel for early warning and inhibition of post-surgical tumor recurrence. a** To construct a postoperative embedded hydrogel with integrated diagnostic and immunotherapy properties, we designed two types of DNA templates. One contains complementary sequences of PDL1 aptamer and CpG, and the other contains complementary sequences of tumor recurrence sensor binding sites. These two types of circular DNA templates and the photosensitizer Ce6-cDNA are used to synthesize light-responsive immunomodulatory hydrogels (CPDH-Ce6) by one-pot rolling circle amplification (RCA) reaction. The circular ATP sensor (cAS) was then loaded onto CPDH-Ce6 by sequence hybridization, resulting in CPDH-Ce6@cAS. **b** After primary tumor resection, CPDH-Ce6@cAS was locally embedded. Fluorescence recovery of cAS indicated tumor recurrence. Under external laser irradiation, CPDH-Ce6@cAS exerted photodynamic therapy and generated ROS to induce the hydrogel to release PDL1 aptamer and CpG. **c** The released CpG and TAA jointly promote DC maturation and T cell activation, and AptPDL1 blocks the binding of PD1 and PDL1 proteins to inhibit immune checkpoints and promote immune pathways to kill tumor cells.

photodynamic immunotherapy to prevent tumor recurrence and metastasis.

## Results

### Synthesis and characterization of CPDH-Ce6@cAS

We developed a multifunctional DNA hydrogel (CPDH) formed from two partially hybridized circular DNA templates encoding CpG and PDL1 aptamers by a one-pot rolling circle amplification (RCA) reaction (Supplementary Table S1). As the agarose gel electrophoresis results showed (Fig. 2a), DNA template-1 and DNA template-2 were reacted with primers and ligated separately. After a one-pot RCA reaction, the morphology of the CPDH was imaged using cryo-scanning electron microscopy (cryo-SEM) and fluorescence microscopy imaging (Fig. 2b, Supplementary Fig. 1). The results presented that CPDH showed a microscopic reticulation. Rheological studies showed that the CPDH had a high storage modulus (G′) compared to the loss modulus (G″), indicating the formation of a soft hydrogel (Fig. 2c).

To characterize the distribution of Ce6 in the hydrogels, DNA hydrogels were stained with the DNA-specific dye SYBR Green II and Ce6 itself exhibited red fluorescence. As demonstrated by confocal imaging, the signal from Ce6 was uniformly distributed in the hydrogel and colocalized with the DNA backbone, indicating successful encapsulation of Ce6 in the CPDH (Fig. 2d). In addition, ultraviolet/visible (UV/vis) spectra showed that both Ce6-cDNA and CPDH-Ce6 have characteristic absorption peaks (the broad peak at 667 nm, Fig. 2e), respectively, further indicating loading of Ce6 on the CPDH. The ROS generation capacity of CPDH-Ce6 was characterized by electron spin resonance (ESR) (Fig. 2f)[44,45]. The Ce6-cDNA sample produced a $^1O_2$ characteristic peak with a 1:1:1 triple signal, while CPDH did not exhibit any signal. The CPDH-Ce6 sample generated less $^1O_2$ than Ce6-cDNA, likely due to some of the generated $^1O_2$ being scavenged by breaking DNA strands in the gel. In addition, the light-generated ROS of CPDH-Ce6 was confirmed by a commercial singlet oxygen fluorescence probe (SOSG). Supplementary Fig. 2a showed that the fluorescence of the CPDH-Ce6 group increased gradually within 10 min after laser irradiation, suggesting the generation of ROS was dependent on the irradiation time.

To examine the loading capacity of CPDH with Ce6, we first determined the optimal time to introduce Ce6-cDNA into the RCA reaction (0 h, 1 h, 2 h, 4 h, 6 h, 8 h). Specifically, we tested various time points (0 h, 1 h, 2 h, 4 h, 6 h, 8 h) to determine the time at which the highest loading efficiency was achieved. The results indicated that the highest loading efficiency was achieved when Ce6-cDNA was added after 2 h of the RCA reaction (Supplementary Fig. 3a, b). Additionally, we observed that adding Ce6-cDNA at 0 h and 1 h significantly impacted the formation of hydrogels, whereas adding Ce6-cDNA at different time points after 2 h (2 h, 4 h, 6 h, 8 h) did not significantly interfere with the formation of DNA hydrogels. Therefore, we selected the time point of 2 h in the RCA reaction to add Ce6-cDNA for the synthesis of DNA hydrogels (Supplementary Fig. 4).

We then incubated CPDH (0.2 μM circular template) with different concentrations of Ce6-cDNA (1 μM, 2 μM, 4 μM, 8 μM, and 16 μM) and evaluated its loading capacity by measuring the fluorescence intensity Ce6 on CPDH. As shown in Supplementary Figure 3c, the loading of Ce6 on CPDH gradually increased with higher Ce6-cDNA concentration and reached a plateau at approximately 8 μM. These results indicate that CPDH loaded Ce6 at a ratio of 1:40 to form CPDH-Ce6, with a loading efficiency of 62.5%. Notably, the optimal loading efficiency was achieved at a concentration of 6 μM Ce6 (1:20), with an efficiency of about 83.4%. For subsequent experiments, we selected the 1:20 ratio of Ce6-cDNA to CPDH to synthesize the integrated DNA hydrogel.

To confer in situ warning of recurrence with DNA hydrogels, a detection module (ATP sensor) was modified on CPDH by nucleic acid hybridization. To improve the biostability of ATP sensor in vivo, we

designed a bicyclic ATP sensor. It is modified with a fluorophore (Cy5) and a quencher (BHQ2) to enable Förster resonance energy transfer (FRET)-based detection. As shown in Fig. 2g and Supplementary Fig. 5a, the successful preparation of cAS was verified by polyacrylamide gel electrophoresis (PAGE). The nuclease resistance of cAS was then assessed by incubating them with combined exonucleases Exo I (0.15 U/μL) and Exo III (0.5 U/μL). By agarose gel electrophoresis analysis, the linear ATP sensor (lAS) was completely degraded by Exo I and Exo III within a short time. In contrast, cAS resistant to enzymes and maintained its integrity after 24 h (Supplementary Fig. 5b), verifying the enhanced biostability of cAS.

We further evaluated whether cAS could respond to ATP effectively. The cAS was incubated with different concentrations of ATP. The fluorescence of Cy5 gradually recovered with increasing ATP levels, indicating that the competitive binding of ATP to its specific site triggers the dehybridization of the bicyclic structure (Fig. 2h, i). Notably, the extracellular ATP concentration of TME ($100–500 \times 10^{-6}$ M) is much higher than that of normal tissues ($10–100 \times 10^{-9}$ M) and fits well within the detection window ($0–1000 \times 10^{-6}$ M)[46,47]. As a control, some crucial nucleotides in the ATP aptamer sequence were mutated to obtain crAS, and no ATP response was seen in the control group (Supplementary Fig. 2b), confirming the critical role of the aptamer sequence in sensor design.

After the detection module produces an early warning signal, laser irradiation was given to trigger PDT and induce the disassembly of hydrogel to release CpG and PDL1 aptamers. We first analyzed the morphology of CPDH-Ce6 under laser irradiation by cryo-SEM. The results showed that the microscopic network of CPDH obviously collapsed after laser irradiation (Fig. 2j). To estimate the release ratio, we assembled FAM-labeled cDNA on CPDH-Ce6 and detected the FAM fluorescence in the supernatant after different laser irradiation time points. With light exposure (10 min per day, 5 days), increasing cDNA-FAM accumulated in the solution, and the cumulative release of cDNA-FAM in the solution elevated by 2.1-fold in the light irradiated group compared to the control group (Fig. 2k). The appearance of CPDH-Ce6 and agarose gel electrophoresis of the solution also supported the idea that light irradiation promotes DNA hydrogel self-disassembly (Supplementary Fig. 6).

Since the ROS generated by PDT doesn't have a sequence specificity for DNA strand breakage[43,48,49], it is necessary to ensure that functional CpG and PDL1 aptamers are released during light exposure. We employed a fluorescence-based molecular beacon assay to analyze functional CpG and PDL1 sequences exposed in solution. The results showed (Fig. 2l) that the amount of CpG and PDL1 aptamer sequences exposed in solution significantly increased after laser irradiation, which was 3.2 and 5.2-folds more than the control group, respectively. Furthermore, to evaluate the proportion of intact and functional PDL1 aptamer and CpG sequences that were released from DNA hydrogels, molecular beacons were incubated with laser-irradiated CPDH-Ce6 hydrogels synthesized by 10 μM primer/circle hybridization at 37 °C for 30 min. We calculated the release rate using long single-stranded DNA synthesized by 10 μM primer/circle hybridization as a control. The release rates for the functional PDL1 aptamer and CpG sequences were identified as 70.1% and 63.9%, respectively. This indicates that irradiation induced the release of the majority of the functional sequences from CPDH-Ce6. In addition, gene sequencing was performed to identify CpG and the integrity of the PDL1 aptamer sequence (Fig. 2m), suggesting that photo-activated self-degrading CPDH could boost the release of CpG and PDL1 aptamers.

### In vitro tumor cell capture and ATP detection using CPDH-Ce6@cAS

CPDH-Ce6@cAS was designed to contain PDL1 aptamer for the capture of tumor cells and achieves timely detection (Fig. 3a). Nucleic acid aptamers offer several advantages over antibodies, including large-

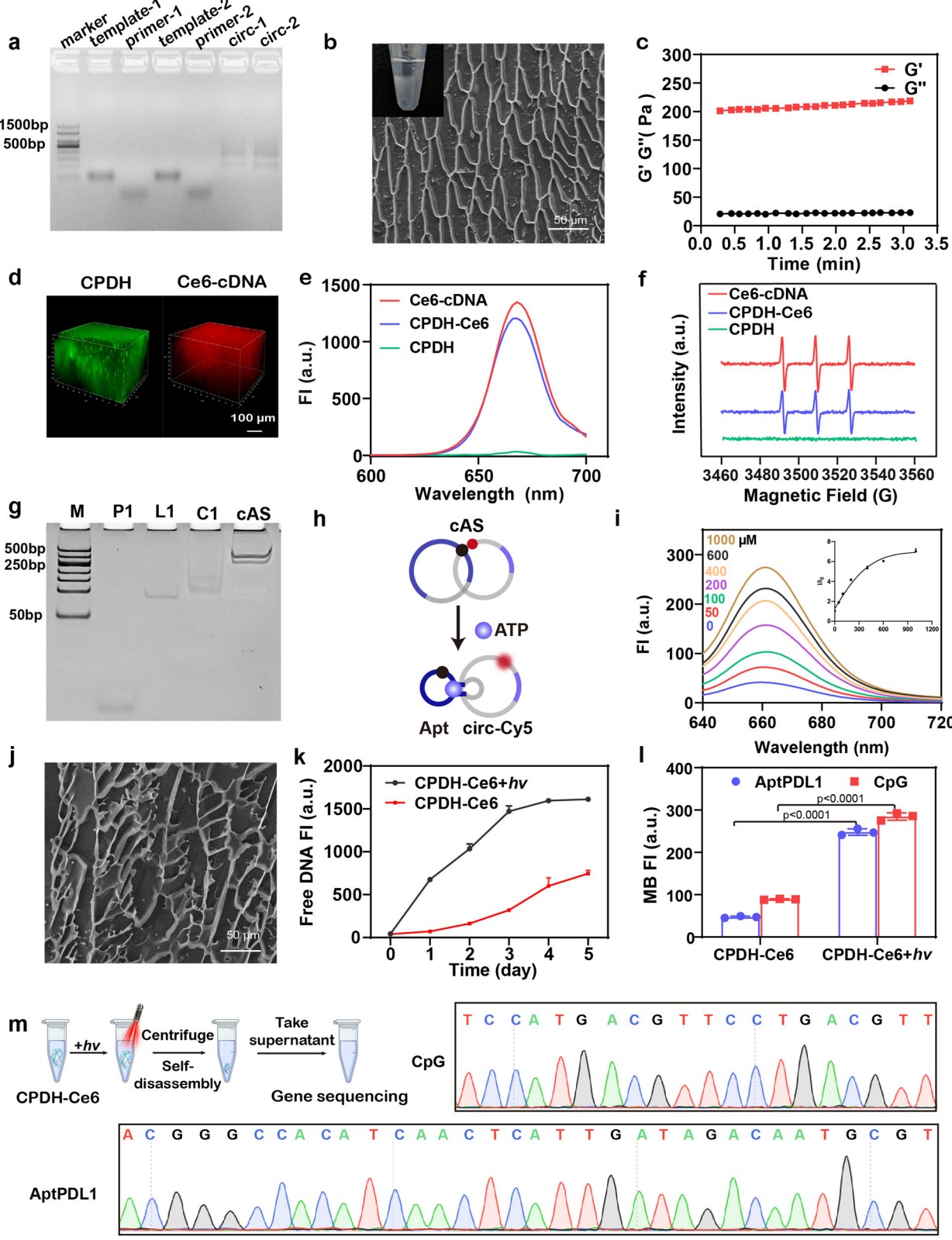

scale synthesis, ease of modification, and low immunogenicity[50–52]. Importantly, the programmable nature of nucleic acid aptamers enables the PDL1 aptamer to be encoded in DNA hydrogel systems. This allows the DNA hydrogel to specifically bind the PDL1 protein overexpressed on the surface of tumor cells, facilitating in situ capture and enrichment of tumor cells[53–55]. Besides, we examined the expression levels of PDL1 protein in different tumor cells by Western-blot,

which demonstrated that B16F10 cells (mouse melanoma hypermetastatic cells) have relatively high PDL1 expression (Supplementary Fig. 8). As a result of these features, we selected the PDL1 aptamer as the cell enrichment module for our study.

To test the tumor cell capture ability of CPDH, SYBR Green II-labeled DNA hydrogels with or without the encoded PDL1 aptamer (CDH and CPDH) were co-incubated with Cell Mask™ Deep Red Plasma

**Fig. 2 | Synthesis and characterization of CPDH-Ce6@cAS. a** 2% agarose gel electrophoresis of template-1, primer-1, template-2, primer-2, circ-1 and circ-2. The experiments were repeated three times independently. **b** Morphology of CPDH observed by cryo-SEM. Scale bar: 50 µm. The experiments were repeated three times independently. **c** Rheological tests of the CPDH. (G': Storage modulus; G": loss modulus). **d** 3D stacking image showing CPDH and Ce6-cDNA. **e** Fluorescence spectra of Ce6-cDNA, CPDH, and CPDH-Ce6. **f** Electron spin resonance (ESR) spectroscopy of $^1O_2$ generation by Ce6-cDNA, CPDH, or CPDH-Ce6. $^1O_2$ was trapped by TEMP after 660 nm laser irradiation (0.2 Wcm$^{-2}$) for 10 min. **g** Characterization of circular ATP sensor using Native PAGE (12%). **h** Schematic diagram of the ATP-induced de-hybridization of the circular structure formed by the BHQ2-modified Apt and Cy5-modified hybrid chains. **i** Fluorescence spectra showing the fluorescence recovery of cAS-Cy5 loaded on DNA hydrogels in response to different concentrations of ATP. **j** Cryo-SEM imaging of CPDH-Ce6 with laser irradiation. The experiments were repeated three times independently. **k** Fluorescence of free nucleic acid fragments released in the supernatant of CPDH-Ce6 (with or without 660 nm laser irradiation). ($n = 3$ independent experiments). **l** Fluorescence-based molecular beacon test to estimate the integrity and accessibility of functional AptPDL1 and CpG released by CPDH-Ce6 (with or without 10 min laser irradiation). Data are analyzed by a two-sided Student's $t$-test and shown as mean ± SD ($n = 3$ independent experiments). **m** Characterization of the integrity of PDL1 aptamer and CpG sequences released by irradiated CPDH by gene sequencing. Source data from (**c**, **e**, **f**, **i**, **k,** and **l**) are provided as a Source Data file.

membrane Stain-stained B16F10 cells. The CDH and CPDH hydrogels were placed into tubes, followed by the addition of 100 µL cell culture medium containing variable numbers of tumor cells, then incubated at 37 °C for 1 h, washed gently twice with PBS, and subsequently imaged by IVIS Spectrum. The results showed that PDL1 aptamer played a crucial role in the capture and enrichment of tumor cells in DNA hydrogel (Fig. 3b, c). With the increasing amount of CPDH, the capture rate of tumor cells rose from 16.3% to 65%, while the corresponding capture rate in CDH did not change significantly (Fig. 3g), indicating that the PDL1 aptamer encoded in the hydrogel can efficiently capture tumor cells. As the number of tumor cells captured by CPDH-Ce6@cAS increases, rising local ATP concentrations was supposed to trigger the cAS to generate fluorescent signals.

To assess the sensitivity of cAS to monitor tumor recurrence, we treated CPDH-Ce6@cAS in test tubes with different concentrations of B16F10 cells and imaged the fluorophore signal recovered by cAS dehybridization. The experimental results showed that the intensity of the fluorescence signal detected by the ATP sensor enhanced with increasing cell count in the density range of 0–5000 cells/µL, indicating its high sensitivity in detecting tumor recurrence (Fig. 3d, e). Furthermore, the fluorescence signal detected by the ATP sensor reached a plateau at a cell count of 500 cells/µL and had a linear correlation ($Y = 66.12X + 4432$, $R^2 = 0.9660$) within a certain range of tumor cell concentrations (0–50 cell/µL) (Fig. 3f). The results of in vitro experiments showed that cAS has a high sensitivity for local tumor cell detection, providing a basis for early warning of tumor recurrence in vivo.

To determine the effect of PDT on the viability of captured tumor cells, CPDH-captured tumor cells were treated with laser irradiation for 10 min and then slightly degraded by the digestive enzyme DNase I. The released tumor cells were continued to be cultured for 14 days, stained with crystal violet, and the formation of cell clones was observed by microscopy (Fig. 3h). The cell cloning rate was quantified by image-J. As indicated in Fig. 3i, the cell cloning rate in the laser-treated group was significantly reduced, which was 49.45% of the non-treated group. Together, these data indicate that the tumor cell capture ability of CPDH-Ce6@cAS could facilitate ATP detection and PDT therapy.

## In vivo monitoring of tumor recurrence via post-operative encapsulated hydrogels

After verifying the sensing activity of cAS at the cellular level, we then explored whether postoperatively embedded CPDH@cAS could monitor tumor recurrence in time by the mechanism of local tumor cell enrichment and ATP-responsive fluorescent "off-on" signals. We first examined the stability of cAS for in vivo detection of tumor recurrence. CPDH@cAS and CPDH@lAS were constructed and embedded into the partially excised tumor in situ of mice and fluorescence changes were recorded at different time points. At 48 h after surgical implantation, the fluorescent signal retained in the cAS group was 1.6 times higher than that in the lAS group (Supplementary Fig. 9), indicating that cAS was more stable in vivo than lAS. The prolonged

residence time of cAS may be due to the absence of 3' and 5' phosphodiester bonds, which avoids the degradation of exonuclease in the TME[56,57].

In addition, the stability of the hydrogel system is the basis for the detection and treatment of tumor recurrence in the post-operative period. We next assessed DNA hydrogel stability in vitro and in vivo. As shown in Supplementary Fig. 10a, we simulated a physiological environment in vitro with 10% FBS and found that the DNA hydrogels could be stored morphologically stable for 10 days. We encapsulated fluorescent group Cy5-labelled Cy5-cDNA and CPDH-Ce6-Cy5 at the surgical site in vivo. As shown in Supplementary Fig. 10b, the fluorescence of Cy5-cDNA almost disappeared after about 1 day in mice, whereas the fluorescence intensity of DNA hydrogels was still detectable after 6 days, indicating that the DNA hydrogels can remain stable in vitro and in vivo for about 1 week. The long-term stability of DNA hydrogels may be further improved by combining them with polymers or transitioning to L-nucleic acid scaffolds[58,59].

To examine the ability of implanted DNA hydrogel for retaining small number of tumor cells within the local environment, we constructed a postoperative model by introducing a specific number of tumor cells into the completely resected surgical area (Fig. 4b and Supplementary Fig. 11). The experimental results demonstrated that CPDH@cAS (with PDL1 aptamers) exhibited strong co-localization with tumor cells compared to CDH@cAS (without PDL1 aptamers), suggesting that adding the PDL1 aptamer in the DNA hydrogel system facilitates the in vivo capture and enrichment of tumor cells.

Since the concentration of ATP in tumor microenvironment was much higher than that of normal tissues[60,61], we then tested whether cAS could specifically detect elevated ATP at tumor sites. Specifically, three groups of mice with B16F10 tumors on their right armpit were embedded with CPDH@crAS, CDH@cAS, and CPDH@cAS after surgery, respectively. We simulated tumor recurrence by surgical incomplete resection (Fig. 4a). Over the time course, the CPDH@cAS group in mice with recurrent tumors showed a strong fluorescence signal in situ and rapidly achieved a maximum at approximately 1 h after implantation. In contrast, due to the absence of tumor cell enrichment, fluorescence in the CDH@cAS group was slower to reach a plateau and remained at a lower level than in the CPDH@cAS group. As a control with no response to ATP, CPDH@crAS did not produce significant fluorescence at the tumor site (Fig. 4c). Quantitative analysis indicated that CPDH@cAS produced signals approximately 1.9 and 3.3-fold higher than that of CDH@cAS and CPDH@crAS at 1 h post-injection, respectively (Fig. 4d). Since the hydrogel was labeled with Cy5, even quenched with BHQ2, there would still be some background signal[46,62]. Imaging analysis was performed to reduce the impact on data readout by reducing and stabilizing the baseline of the monitoring signal.

After 48 h of hydrogel implantation, tumors and major organs of mice were harvested for ex vivo imaging (Fig. 4e). The intratumoral fluorescence intensity in the CPDH@cAS group was 1.6- and 2.5-fold higher than in the groups receiving CDH@cAS and CPDH@crAS, respectively, with significant differences. And there was no non-

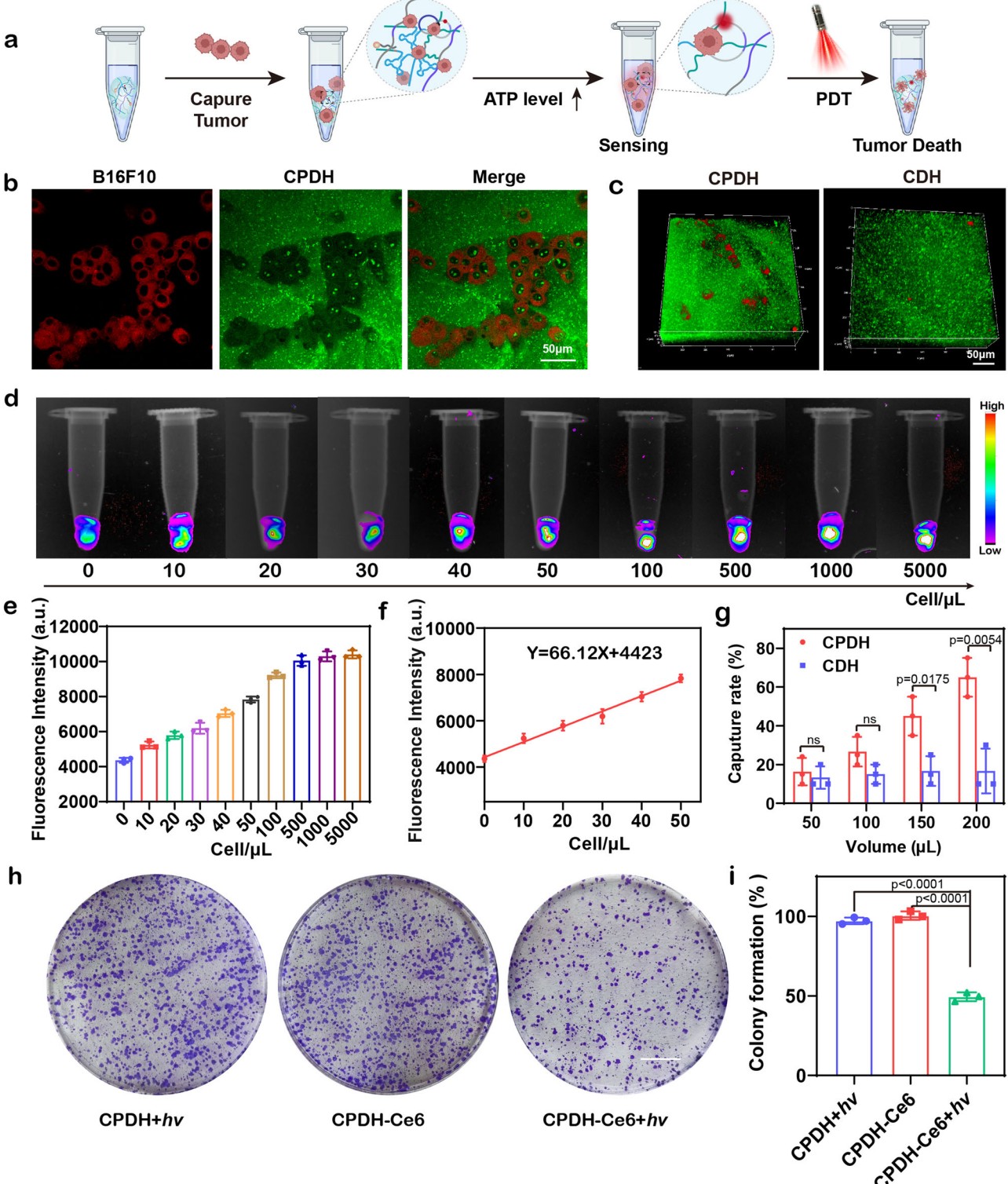

**Fig. 3 | Tumor cell capture of CPDH-Ce6@cAS to facilitate ATP detection and PDT therapy in vitro. a** Schematic diagram of CPDH-Ce6@cAS capturing tumor cells and then killing them by PDT. **b** Confocal microscopy imaging of B16F10 cells (stained with Cell Mask™ Deep Red) was captured in CPDH-Ce6 (stained with SYBR Green II). Scale bar: 50 μm. **c** 3D imaging of B16F10 cells captured by hydrogels with or without PDL1 aptamers. Scale bar: 50 μm. CPDH@cAS was incubated with different numbers of B16F10 cells, and **d** the ATP sensor produced fluorescent images of the detected signal and **e** the corresponding fluorescence intensity. **f** Linear relationship of the fluorescence signal detected by the ATP sensor at 0–50 cells/μL. **g** Relationship between cell capture rate and volume of CPDH-Ce6 (with or without PDL1 aptamer) Data are analyzed by two-sided Student's $t$-test and shown as mean ± SD ($n$ = 3 independent experiments). **h** Images of clone formation and **i)** corresponding clone formation rates of B16F10 cells after CPDH+$hv$, CPDH-Ce6, and CPDH-Ce6+$hv$ treatment. Scale bar: 1 cm. Data are mean ± SD ($n$ = 3 independent experiments). Data are analyzed by two-sided Student's $t$-test. Source data from (**e, f, g**, and **i**) are provided as a Source Data file.

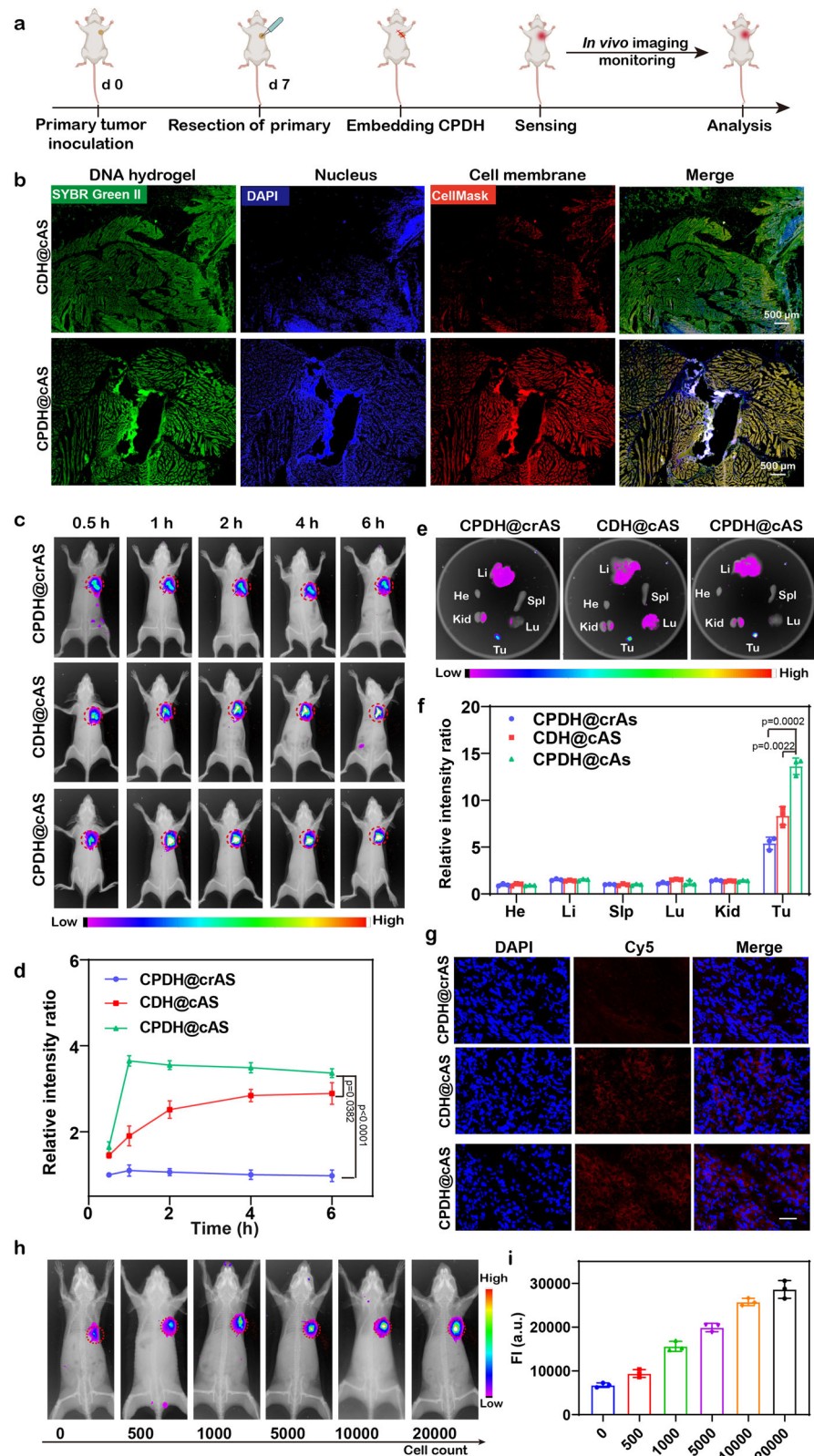

specific signal in the major organs (heart, liver, spleen, lung, and kidney) in all three groups, demonstrating that the hydrogel system could be stable and won't release its fluorophore to other organs after being implanted for 48 h (Fig. 4f). Moreover, fluorescence imaging of tumor sections revealed that CPDH@cAS produced stronger fluorescence in the tumor tissue compared to the other two controls (Fig. 4g, Supplementary Figure 12).

Furthermore, we conducted an analysis on the sensitivity of CPDH@cAS for in vivo tumor cell detection by administering varying numbers of tumor cells (0, 500, 1000, 5000, 10000, and 20000 cells count) at the site of fully resected surgery (Supplementary Figure 13). The obtained experimental results demonstrated a positive correlation between the fluorescence generated by the ATP sensor and the number of tumor cells, highlighting the high sensitivity of the sensor in

**Fig. 4 | CPDH@cAS for tumor recurrence detection. a** Schematic diagram of detecting tumor recurrence in vivo. **b** Frozen section fluorescence imaging of DNA hydrogels (with or without PDL1 aptamer) capturing tumor cells in vivo. Green: DNA hydrogel, Blue: cell nucleus, Red: tumor cell membrane. Scale bar: 500 μm. The experiments were repeated three times independently. **c** Fluorescence images of mice with tumor recurrence after embedding different hydrogel sensors. Tumors are indicated by red circles. **d** Relative fluorescence quantification of mice tumors at different time points ($n = 3$ independent experiments). Data are analyzed by two-sided Student's $t$-test and shown as mean ± SD. **e** Fluorescence images of isolated tumors and main organs harvested 48 h after embedding different hydrogel. **f** Relative fluorescence intensity quantification of tumors and main organs in (**e**). Data are analyzed by two-sided Student's $t$-test and shown as mean ± SD. **g** Fluorescence images of ATP sensors from tumor sections of differently treated mice. Scale bar: 50 μm. **h** Fluorescence map of CPDH@cAS at 1 h with different densities of tumor cells. **i** CPDH@cAS detected the signal intensity of different densities of tumor cells at 1 h. Date are presented as mean ± SD ($n = 3$ independent experiments) and analyzed by two-sided Student's $t$-test. Source data from (**d, f,** and **i**) are provided as a Source Data file.

detecting tumor recurrence. In addition, our findings revealed that the ATP sensor is capable of responding rapidly to the relapse of tumor cells and reaches its maximum value at 1 h, emphasizing its timely nature (Fig. 4h, i). Taken together, these results demonstrated that CPDH@cAS allows spatially selective visualization of early tumor recurrence and improves the persistence and reliability of warning signals.

## Immunotherapeutic hydrogel for tumor recurrence inhibition and distant tumor treatment

To validate the antitumor therapeutic effect of CPDH-Ce6 in vivo, we used an incompletely resected luciferase-labeled B16F10 (Luc-B16F10) tumor model to simulate tumor recurrence[63]. The Luc-B16F10 cancer cells were seeded in the contralateral side of the primary tumor to mimic distant tumor. When the size of the tumors reached 100 mm³, the primary tumor was partially excised and hydrogels were implanted into the tumor resection cavity followed by daily laser irradiation for 5 days (Supplementary Fig. 14, Fig. 5a). The untreated group served as a control, bioluminescence signals from B16F10 cancer cells were used to monitor tumor growth (Fig. 5b).

According to the results, CPDH-Ce6 treatment significantly inhibited local tumor recurrence and reduced distal tumor growth compared to the untreated groups (Fig. 5c, d, Supplementary Fig. 15), and the body weights of mice were not affected by the treatment (Supplementary Fig. 16a). The tumor bioluminescence intensity and tumor volume of the mice in this group were the lowest on the 17th day, calculated from the tumor bioluminescence intensity of the pairs of mice. Calculation formula of tumor inhibition rate: TIR = (1 − $BI_{TI}$/ $BI_{T0}$) × 100% (TIR: tumor inhibition rate, $BI_{TI}$: tumor bioluminescence intensity in the treatment group, $BI_{T0}$: tumor bioluminescence intensity in the control group.). In addition, we calculated the inhibition rates of primary and distal tumors in different groups of mice by the formula in Supplementary Table S4. The inhibition rates of primary and distal tumors were about 88.1% and 56.3%, respectively (Fig. 5e, f). In contrast, DH-Ce6 (without DNA-based immunomodulators) showed lower tumor suppression rates on both sides (about 64.7% and 45.7%, respectively), suggesting that PDL1 aptamers and CpG play an essential role in the treatment.

Moreover, hematoxylin and eosin (H&E) staining of the major organs collected from mice after the treatment procedure (Supplementary Fig. 16b) and routine blood analysis (Supplementary Fig. 16c) showed no obvious abnormalities, indicating that DNA hydrogels did not induce marked side effects to mice. Furthermore, after laser irradiation, massive ROS was noted in the tumor tissues of the mouse in the CPDH-Ce6 group and promoted the deep penetration of Ce6 in the tumor tissues (Fig. 5g, h). The terminal deoxynucleotidyl transferase-mediated dUTP-biotin nick end labeling (TUNEL) and H&E staining of tumor sections showed that the irradiated CPDH-Ce6 group induced substantial necrosis and apoptosis in primary and distant tumors (Fig. 5i).

## Photoactivated DNA hydrogel-induced immune response in vivo

The specific immune response triggered at the tumor site was then investigated. Specifically, the primary tumor tissue was excised and DNA hydrogel was embedded in situ, and light exposure was performed for 10 min daily, while the distal tumor was left untreated. The residual tumors were harvested 4 days postoperatively and analyzed using immunofluorescence staining and flow cytometry (the gating strategy is shown in Supplementary Fig. 17). As shown in Fig. 6a, remarkable CRT exposure and HMGB1 release in primary tumors of CPDH-Ce6+$hv$-treated mice suggested an effective immunogenic cell death (ICD) activation. On the other hand, the percentage of tumors-infiltrating CD80⁺CD86⁺ DCs in local and distant tumors treated with CPDH-Ce6+$hv$ was 3.8-fold and 2.8-fold higher than that of the untreated group, respectively, indicating that CPDH-Ce6+$hv$ could effectively promote DC maturation (Fig. 6b, Supplementary Figure 18a, b). In contrast, the percentage of DC activation in the CPDH-Ce6 and DH-Ce6+$hv$ treated groups were lower, implying that the synergistic treatment with PDT and immune activation may contribute to the high immunotherapeutic efficiency.

The local presentation of tumor antigens by DC cells could contribute to the increase of CD8⁺ T cells, which triggers systemic antitumor immunity. Consistent with the treatment results, we detected 3.1-fold and 2.2-fold more tumor-infiltrating CD3⁺CD8⁺ T cells in a local and distant tumor in the CPDH-Ce6+$hv$ group than in CPDH-Ce6 and DH-Ce6+$hv$ groups, respectively (Fig. 6c, Supplementary Fig. 18c). Meanwhile, the proportion of Tregs (Foxp3⁺ T cells) decreased, indicating that CPDH-Ce6+$hv$ could also reduce tumor-associated immunosuppression (Fig. 6e, Supplementary Fig. 19a). Immunofluorescence staining visually demonstrated a dramatic increase in CD80⁺CD86⁺ DC cells, CD4⁺ T cells, and CD8⁺ T cells in residual tumors after CPDH-Ce6+$hv$ treatment, consistent with the results of flow cytometry (Fig. 6d).

The PDL1 protein is typically expressed at low levels in healthy tissues but becomes upregulated in tumor microenvironments, rendering it a promising target for immune checkpoint inhibition therapies[54,64]. Upon binding to PDL1 on the tumor cell surface, the PDL1 aptamer may interfere with PDL1 protein function, leading to compensatory overexpression of PDL1 by the tumor cells and ultimately reinforcing the anti-tumor immune response[36]. (Supplementary Fig. 19b, c). Secretion of cytokines such as IFN-γ, IL-6, and TNF-α further confirmed the effective tumor-specific immune response induced by CPDH-Ce6+$hv$ treatment (Fig. 6f). In conclusion, these data suggested that the designed photodynamic immunomodulatory DNA hydrogels can trigger an effective immune response, resulting in effective postoperative tumor suppression.

## Evaluation of DNA hydrogel for suppression of lung metastases after surgical resection of primary tumors

Lung metastasis is another major clinical problem in malignant cancer patients after surgery. To evaluate the effect of DNA hydrogel treatment on lung metastases after surgical resection of primary tumors, we constructed and treated a standard B16F10 lung metastasis model (Fig. 7a). Then, lung tissue was collected from each group of mice for analysis and comparison. Lung metastases were assessed by photographs and histological analysis of H&E staining. CPDH-Ce6+$hv$-treated mice showed much less metastatic nodules in the lungs compared to control and other groups, suggesting that the DNA hydrogel could effectively prevent postoperative lung metastases (Fig. 7b, c). H&E staining results showed a consistent tendency (Fig. 7d, Supplementary Fig. 20).

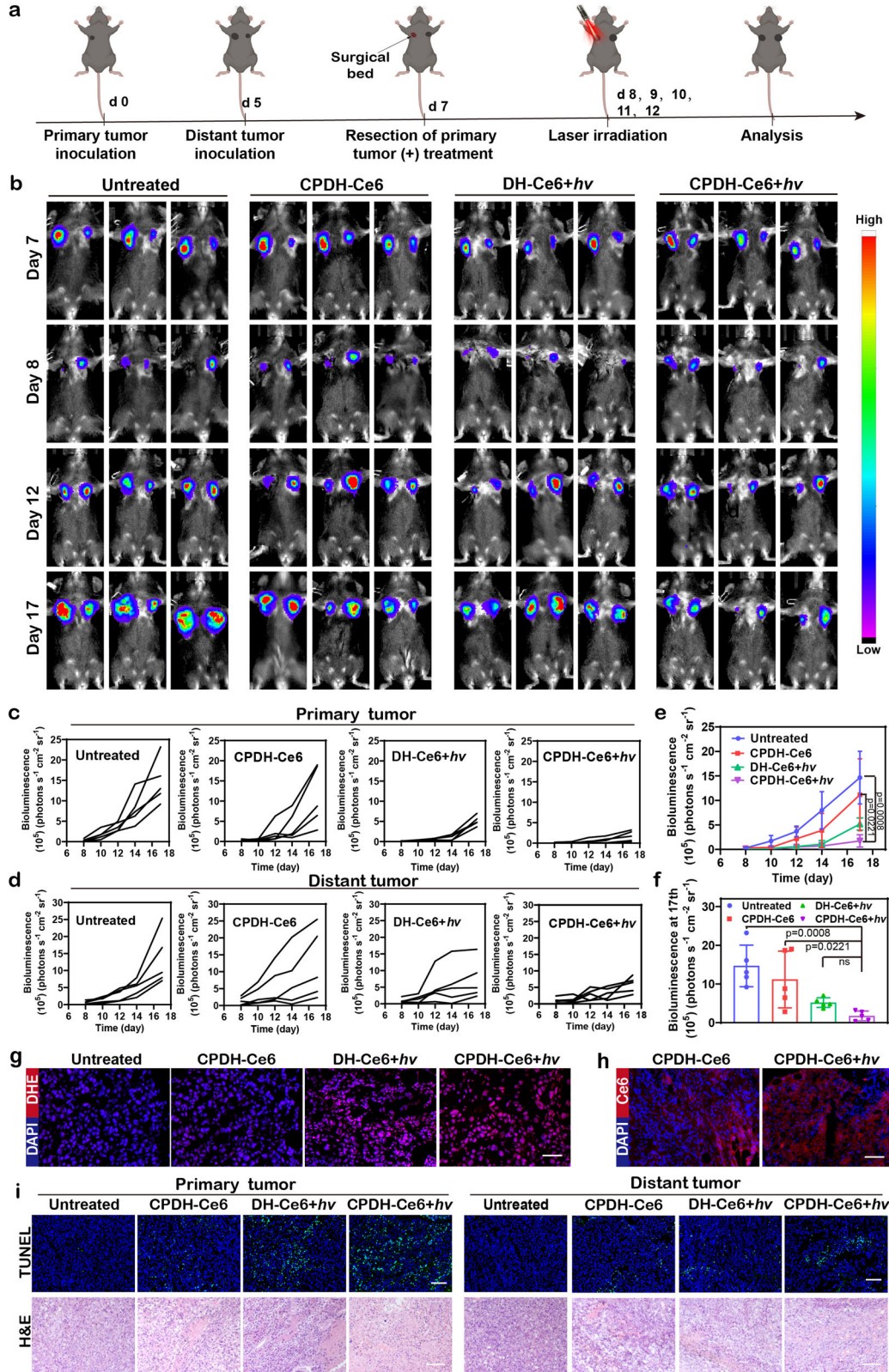

To assess the recruitment of immune cells in the lung, we stained frozen lung sections by immunofluorescence followed by CLSM imaging. As shown in Fig. 7e and Supplementary Fig. 21, the proportion of CD3+ T cells, CD8+ T cells, CD80+ DC cells, and CD86+ DC cells was significantly increased in the lung tissue of CPDH-Ce6+*hv*-treated mice, which was 1.7-fold, 6.9-fold, 2.1-fold, and 2.8-fold higher, respectively, compared to the untreated group of mice. These findings

suggest that the treatment was effective in stimulating the immune response in the mice's lungs, as indicated by the increased presence of T cells and DC cells.

In addition, we collected axillary and inguinal lymph nodes ipsilateral to the tumor and analyzed immune cell levels to assess the systemic anti-tumor immune response triggered by DNA hydrogel treatment. As shown in Fig. 7f, compared to the untreated and DH-

**Fig. 5 | Antitumor effect of locally encapsulated CPDH-Ce6 on B16F10 bilateral tumor mice. a** Schematic representation of CPDH-Ce6 treatment in the B16F10 melanoma model with incompletely resected and distant tumors. The resected tumors were designated as primary tumors. **b** Representative tumor bioluminescence imaging of B16F10 tumors in response to local CPDH-Ce6 treatment. **c** Region-of-interest analysis of recurrent and **d** distant tumor bioluminescence intensities. **e** The primary tumor bioluminescence intensities curves of mice after various treatments ($n = 5$ mice per group). Data are analyzed by two-sided Student's $t$-test. **f** Comparison of bioluminescence intensity of tumors in different treatment groups on 17th ($n = 5$ mice per group). Data are analyzed by two-sided Student's $t$-test. **g** Fluorescence analysis of ROS generated from tumor sections in different treatment groups. Scale bar: 50 μm. The experiments were repeated three times independently. **h** Fluorescence images of the distribution of Ce6 in mice tumor sections with or without irradiation. Scale bar: 100 μm. The experiments were repeated three times independently. **i** TUNEL and H&E staining of the primary and corresponding distant tumor section. Scale bars: 100 μm. The experiments were repeated three times independently. Dates are presented as mean ± SD. Source data from (**e**, **f**, **d**, and **f**) are provided as a Source Data file.

Ce6+$hv$ groups, the proportion of CD80$^+$CD86$^+$ DC cells in the CPDH-Ce6+$hv$ group increased by 2.9 and 1.3-fold, respectively, indicating that CPDH-Ce6+$hv$ could promote the maturation of DC cells in lymph nodes. The proportion of CD3$^+$CD8$^+$ T cells in the CPDH-Ce6+$hv$ group increased by 4.5 and 1.5-fold, respectively, indicating that CPDH-Ce6+$hv$ could promote the activation of T cells in lymph nodes (Fig. 7g). Moreover, the proportions of CD3$^+$CD8$^+$ T cells and CD80$^+$CD86$^+$ DC cells in the CPDH-Ce6+$hv$ group were increased by 2.6 and 2.1-fold, respectively, compared to the unirradiated CPDH-Ce6 group, indicating that photoactivated CPDH-Ce6 could exert photodynamic immunotherapeutic effects. Taken together, the results suggested that this DNA hydrogel system can induce robust photodynamic therapy and systemic immune response to inhibit lung metastasis.

## Discussion

The surveillance and management of postoperative malignancy recurrence represent a critical and pressing challenge within the realm of oncology. Despite notable advancements in detection and treatment modalities, the absence of early warning signs and treatment specificity remains a significant concern[5,65,66]. Effective identification of tumor recurrence at an early stage is hindered by the limited number of recurrent tumor cells and low concentrations of biomarkers[67,68]. Moreover, the inadequate immunogenicity and immunosuppression at the site of tumor resection contribute to tumor progression, often resulting in missed treatment opportunities as tumors advance to intermediate or advanced stages[69,70]. Consequently, there is an urgent need to develop more sensitive techniques for detecting tumor recurrence and preventing secondary metastasis subsequent to primary tumor resection.

In this work, we developed a post-operative embedded DNA hydrogel for local enrichment of early relapsed tumor cells to achieve timely photodynamic immunotherapy treatment. We found that as the PDL1 aptamer binding site on the hydrogel increased, its capture rate of tumor cells rose from 16.3% to 65%. Moreover, the fluorescence signal of the modified ATP sensor was stronger as the number of cells captured by the hydrogel increased, and there was a linear relationship between the two at a range of tumor cell concentrations (0–50 cells/μL). This provides a new perspective to enhance the detection and treatment of postoperative tumor recurrence by effectively capturing or recruiting recurrent tumor cells.

In addition, this photo-responsive self-disassembled immunomodulatory DNA hydrogel triggered immunotherapy at the surgical site and elicited a systemic immune response in a mice melanoma tumor model, with an inhibition rate of approximately 88.1% against recurrent tumors. These results highlight the potential of spatiotemporally controlled photodynamic immunotherapy as a promising approach to enhance postoperative tumor treatment.

However, it should be noted that partial resection of tumors or injection of a small number of tumor cells at the surgical site cannot fully replicate postoperative tumor recurrence. Additionally, the stability of DNA hydrogels and detection probes in vivo remains relatively low. It is crucial to enhance the stability of these components and ensure their functionality and reliability in vivo to facilitate the broader application of the designed hydrogel system in clinical settings.

Despite these limitations, the in situ embedded DNA hydrogel has demonstrated promising results in the early detection of postoperative tumor recurrence and timely treatment, which can be further expanded by incorporating various capture, detection, and therapy modules.

## Methods
### Materials and apparatus

The oligonucleotides in Tables S1 and Tables S3 were synthesized by Shanghai Sangon Biological Engineering Technology & Services Co., Ltd. (Shanghai, China). The Ce6-cDNA was purchased from TaKaRa (Dalian, China). phi 29 DNA polymerase T4 DNA ligase, Exonuclease I, and Exonuclease III were purchased from Thermo Fisher Technology Co Ltd. (Waltham, MA, USA). DNA markers were obtained from Sangon Biotech (Shanghai) Co., Ltd. The B16F10 cell line (Mice melanocytes), Hela cell line, B16 cell line, 4T1 cell line, 3T3 cell line, and Luc-B16F10 (Luciferase labeling of melanocytes) were provided by the Cell Bank of the Chinese Academy of Science (Shanghai, China). Annexin V-FITC/PI was purchased from Solarbio Science & Technology (Beijing, China). Tumor Dissociation Kit (mouse) was supplied from Miltenyi. PE/Cyanine7 anti-mouse CD45 Antibody (Catalog no. 103 113), Brilliant Violet 421 anti-mouse CD11c Antibody (Catalog no. 117 343), PE anti-mouse CD80 Antibody (Catalog no. 104 707), Brilliant Violet 650 anti-mouse CD86 Antibody (Catalog no. 105 035), FITC anti-mouse CD3 Antibody (Catalog no. 100 203) and APC anti-mouse CD8a Antibody (Catalog no. 100 711) were provided from Biolegend. (San Diego, CA, USA). Anti-mouse Foxp3 Antibody (bs-23074R) and anti-mouse PDL1 Antibody (bs-1103R) were provided from Bioss. ELISA kit was obtained from Jianglai Biotechnology (Shanghai, China). Electron microscope support comes from Shiyanjia Laboratory (www.Shiyanjia.com). The microscopic imaging and living imaging equipment are from the Modern Analysis and Computing Center of Zhengzhou University.

### Cell culture and animal models

B16F10 cells (IM-M002), B16 cells (IM-M001), and 4T1 cells (IM-M017) were grown in RPMI-1640 medium, containing 1% streptomycin/penicillin and 10% fetal bovine serum (FBS) at 37 °C under 5% $CO_2$. The Luc-B16F10 cells (IML-039), Hela cells (IM-H010), and 3T3 cells (IM-M045) were plated in DMEM medium containing 10% FBS and 1% penicillin/streptomycin. The female C57BL/6 mice (5 - 6 weeks, 18 - 22 g) were acquired from Hunan Slaughter Jingda Laboratory Animal Co. All animals were grown at the condition of 25 ± 2 °C and 55% humidity with a 12 h light/dark cycle. The license number is SCXK (Xiang) 2019-0004. Animal experiment protocols were conducted in accordance with the guidelines of the regional Animal Experimentation Ethics Committee and Zhengzhou University. The animal laboratory's accreditation number is 110 322 211 102 955 054.

### Synthesis of the CPDH-Ce6@cAS

Synthesis of DNA hydrogel by One-Pot RCA Reaction. To prepare the circular template, combine 4 μL DNA template (10 μM) and 4 μL primer (10 μM), 4 μL 10 × T4 DNA ligase buffer (50 mM Tris-HCl, 10 mM $MgCl_2$, 10 mM DTT, and 1 mM ATP) with 26 μL ultrapure water and heated at 95 °C for 5 min. Then the temperature was lowered to room temperature with a gradient of 0.1 degrees per second. Using 2 μL T4 DNA

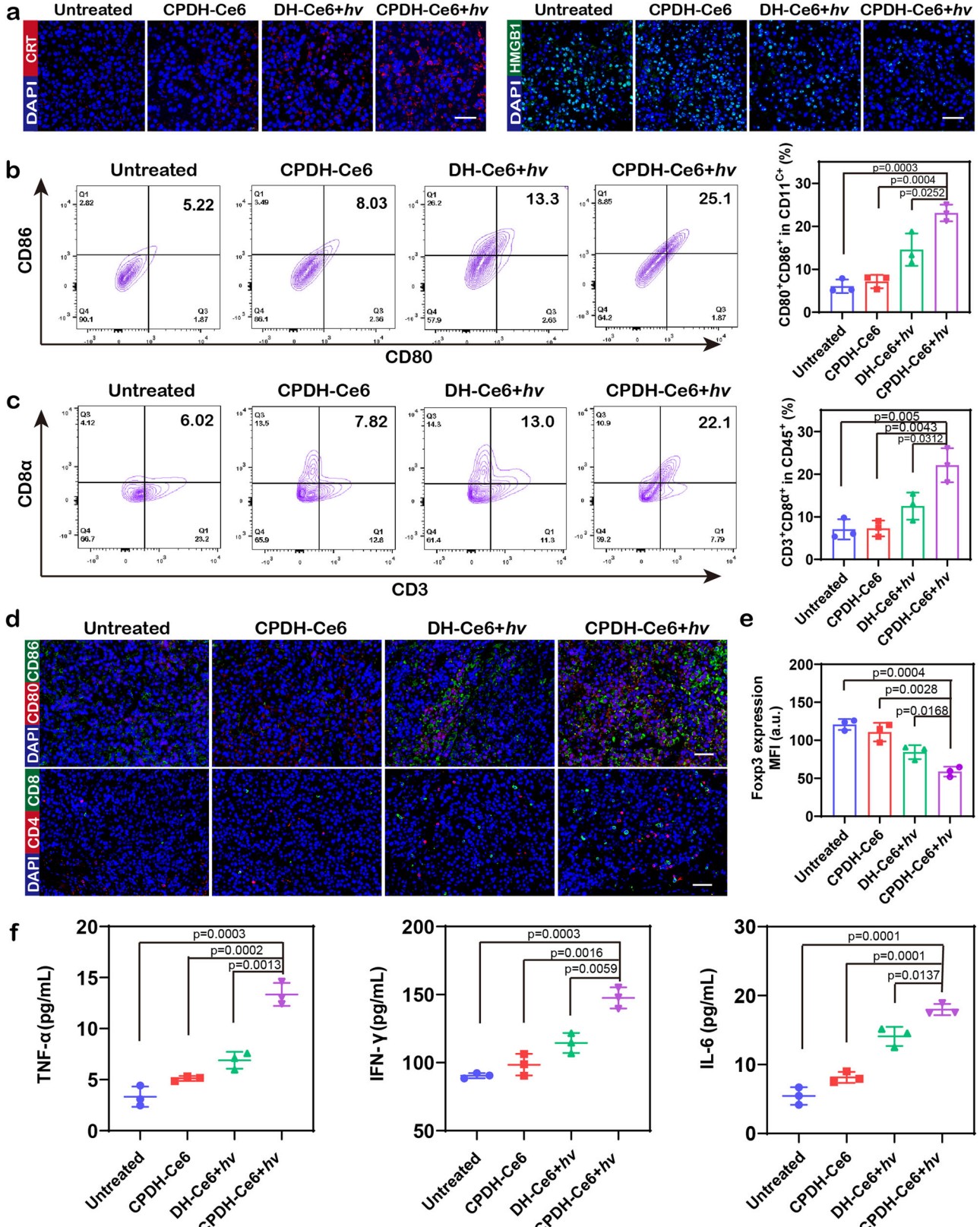

ligase was ligated for 3 h to obtain circ-DNA (circ DNA-1 and circ DNA-2, respectively). Circle-1 (1 μM) and circle-2 (1 μM) with dNTPs (5 mM) and phi 29 DNA Polymerase (0.5 U/μL) were incubated in reaction buffer (33 mM Tris-acetate, 10 mM magnesium acetate, 66 mM potassium acetate, 0.1% (v/v) Tween 20, 1 mM DTT, pH 7.9) at 37 °C for 8 h, 80 °C for 20 min to extinguish the DNA polymerase. The DNA hydrogel was obtained and washed twice with PBS.

For the preparation of Ce6 and cAS-loaded hydrogels (CPDH-Ce6@cAS), CPDH-Ce6 was initially obtained through one-pot rolling circle amplification (RCA) of two partially complementary circular DNA templates and Ce6-cDNA. To prepare the circular ATP sensor (synthetic sequences are shown in Supplementary Table S2 in the Supplementary Information), circular DNA (C1) was first generated using linear DNA (L1) with primers via template-based ligation. The mixture

**Fig. 6 | In vivo synergistic antitumor immune response of CPDH-Ce6.** B16F10 tumors were obtained from mice 4 days after treatment. **a** Immunofluorescence staining of CRT exposure and HMGB1 release in primary B16F10 tumor tissue. Scale bars: 50 μm. The experiments were repeated three times independently. **b** Representative flow cytometric analysis of tumor-infiltrating DCs (CD80$^+$CD86$^+$) and **c** CTLs (CD3$^+$CD8α$^+$), and the quantification results (right) in different groups (*n* = 3 independent experiments). Data are analyzed by two-sided Student's *t*-test and shown as mean ± SD. **d** Immunofluorescence images show DC (red: CD80$^+$,

green: CD86$^+$) and CTLs (red: CD4$^+$ T cell, green: CD8$^+$ T cell) infiltrated in tumors of different groups. Scale bars: 50 μm. **e** The corresponding quantification of the MFI of Foxp3 (*n* = 3 independent experiments). Data are analyzed by two-sided Student's *t*-test and shown as mean ± SD. **f** Immunocytokine levels in sera of mice isolated 4 days after different treatments (*n* = 3 independent experiments). Data are analyzed by two-sided Student's *t*-test and shown as mean ± SD. Source data from (**b**, **c**, **e**, and **f**) are provided as a Source Data file.

was mixed with 1 μL Exo I (20 U/μL) and 0.5 μL Exo III (200 U/μL) at 37 °C for 40 min to obtain circular DNA 1 (C1: 2 μM). Subsequently, all enzymes were inactivated by heating at 80 °C for 20 min. The circular ATP sensor (cAS) was prepared by mixing 20 μL of C1 (2 μM) and 6 μL of 5′-phosphorylated linear DNA 2 (L2: 10 μM) in 1 × T4 DNA ligase buffer was heated at 95 °C for 5 min. Add 2 μL of T4 DNA ligase (5 U/μL) to the mixture and incubate for 3 h at 25 °C. Finally, cAS and CPDH-Ce6 were incubated at 37 °C for 1 h, and CPDH-Ce6@cAS was obtained by nucleic acid strand hybridization.

### Characterization of CPDH-Ce6
The cyclization of DNA templates was verified by mixing 10 μL of DNA template (1 μM) and 2 μL of loading buffer into each sample well via a 2% agarose gel and running at 6 V/cm for 40 min in an ice bath. The results were analyzed with a gel imaging system. Ultraviolet/visible (UV/vis) spectroscopy validated the payload of CPDH-Ce6 on Ce6-cDNA. Specifically, to study the ability of hydrogels to load Ce6, we added different concentrations of Ce6-cDNA to 100 μL of CPDH to obtain CPDH-Ce6. The supernatant in the hydrogel was collected and the fluorescence of Ce6 was measured. The loading of Ce6 by CPDH was calculated by the fluorescence difference.

The mechanical properties of CPDH were tested by a Thermo HAAKE MARS rheometer. The rheology studies were executed on 8 mm parallel plate geometry using 200 μL of DNA hydrogel. The parameters of the test mode were set as follows: frequency set to 1 Hz, strain set to 1%, temperature set to 25 °C, and scan time set to 3 min. In addition, the morphology of the samples was observed by cryo-scanning electron microscopy. Briefly, a large amount of CPDH was placed on the silicon which was plasma cleaned by plasma cleaner (PDC-MG). The samples were frozen in liquid nitrogen and then dried in a freeze-dryer. To observe the microstructure of the hydrogel, CPDH was stained with SYBR Green II and observed by fluorescence microscopy (Nikon Ti-E).

### The effect of the addition of Ce6-cDNA on hydrogel formation during the RCA reaction
Specifically, we tested the effect of adding Ce6-cDNA at six different time points (0 h, 1 h, 2 h, 4 h, 6 h, and 8 h) on the ability of CPDH to load Ce6-cDNA. Firstly, the fluorescence values of Ce6-cDNA were measured by a microplate reader. Next, Ce6-cDNA was added to 100 μL of CPDH solution at different time points of the RCA reaction. At the end of the reaction, centrifugation was performed and the supernatant from each group was added to a 96-well plate. The fluorescence value of Ce6-cDNA in the supernatant of each group was measured. The fluorescence difference was used to calculate the loading of Ce6-cDNA by CPDH.

Next, the effect of adding Ce6-cDNA at different times during the RCA reaction on the morphology of the hydrogels was examined using visual images and cryo-scanning electron microscopy. Specifically, CPDH-Ce6 was obtained by adding Ce6-cDNA at different time points of the RCA reaction. The reaction was carried out at 37 °C and terminated at 80 °C for 20 min to inactivate the polymerase and other active substances. The prepared DNA hydrogels were stained with SYBR Green II for 15 min and washed twice with PBS at the end of staining, then placed under UV light to observe the morphology of the hydrogels formed by adding Ce6-cDNA at different time points. In addition,

the morphology of the DNA hydrogels formed at different time points was observed by cryo-scanning electron microscopy.

### Examine the ATP sensor in vitro in response to ATP
ATP sensor assays were performed by diluting the circular ATP sensor to a concentration of 500 nM and adding ATP at varying concentrations. The circular ATP sensor was incubated with different amounts of ATP for 30 min at 37 °C. Fluorescence spectra were collected at an excitation wavelength of 638 nm.

### Investigate the stability of the circle ATP sensor in vitro
For stability analysis in serum, 2% agarose was performed for characterization. Specifically, the prepared circular ATP sensor and double-stranded ATP sensor were incubated with 0.15 U/μL Exo I and 0.5 U/μL Exo III at 37 °C. At various time points, heat samples at 95 °C to denature proteins in FBS and then store them at −20 °C until analysis. For agarose verification, mix 10 μL of circular ATP sensor or double-stranded ATP sensor sample (1 μM circular ATP sensor or double-stranded ATP sensor) with 2 μL of Loading buffer solution, add to each sample well and place on ice. Run at 6 V/cm for 40 min in the bath. Gel imaging analysis system for viewing gels under UV light at 302 nm.

### Analysis of singlet oxygen ($^1O_2$) production in vitro
The $^1O_2$ generating capacity of CPDH-Ce6 was measured by ESR (JES-FA200). Before detection, each sample solution was illuminated with a 660 nm laser for 10 min. In addition, the production of $^1O_2$ was also assessed using a singlet oxygen green fluorescent probe (SOSG). Briefly: 2 μL of 100 μM SOSG in PBS, CPDH-Ce6, and CPDH respectively, each sample was treated with a 660 nm laser (0.2 Wcm$^{-2}$) irradiated for different times. The fluorescence intensity was measured using a microplate reader.

### Study of Photoactivated degradation of CPDH-Ce6
Firstly, the laser exposure to CPDH-Ce6 for 10 min, and then the photoactivated degradation of CPDH-Ce6 was characterized by agarose gel electrophoresis and SEM. Secondly, the fluorescence-modified cDNA was added to investigate the degradation of CPDH-Ce6. Specifically, the CPDH-Ce6 were treated differently (presence or absence of 10 min laser irradiation), centrifuged, and collected supernatant. The fluorescence density in the supernatant was measured. In addition, a fluorescence-based molecular beacon test (sequence of molecular in shown in Supplementary Table S3) was also designed to estimate the integrity of PDL1 aptamer and CpG sequences in CPDH-Ce6 (giving or not giving 10 min of laser irradiation) solution. After laser exposure, more CpG and PDL1 aptamers were released from the solution, which may open up more molecular beacon probes in the solution to exhibit elevated fluorescence signals. Finally, the integrity of the functional sequences released from the hydrogel was verified by gene sequencing analysis.

### Assessing cell enrichment by hydrogels with or without PDL1 aptamers
To assess the enrichment of cells by PDL1 aptamers, B16F10 cells were to subjected a humidified environment at 37 °C. Following that, cells were collected and counted on a counting plate before being used in the next step. B16F10 cells were stained with 10 μM CellMask™ in Deep

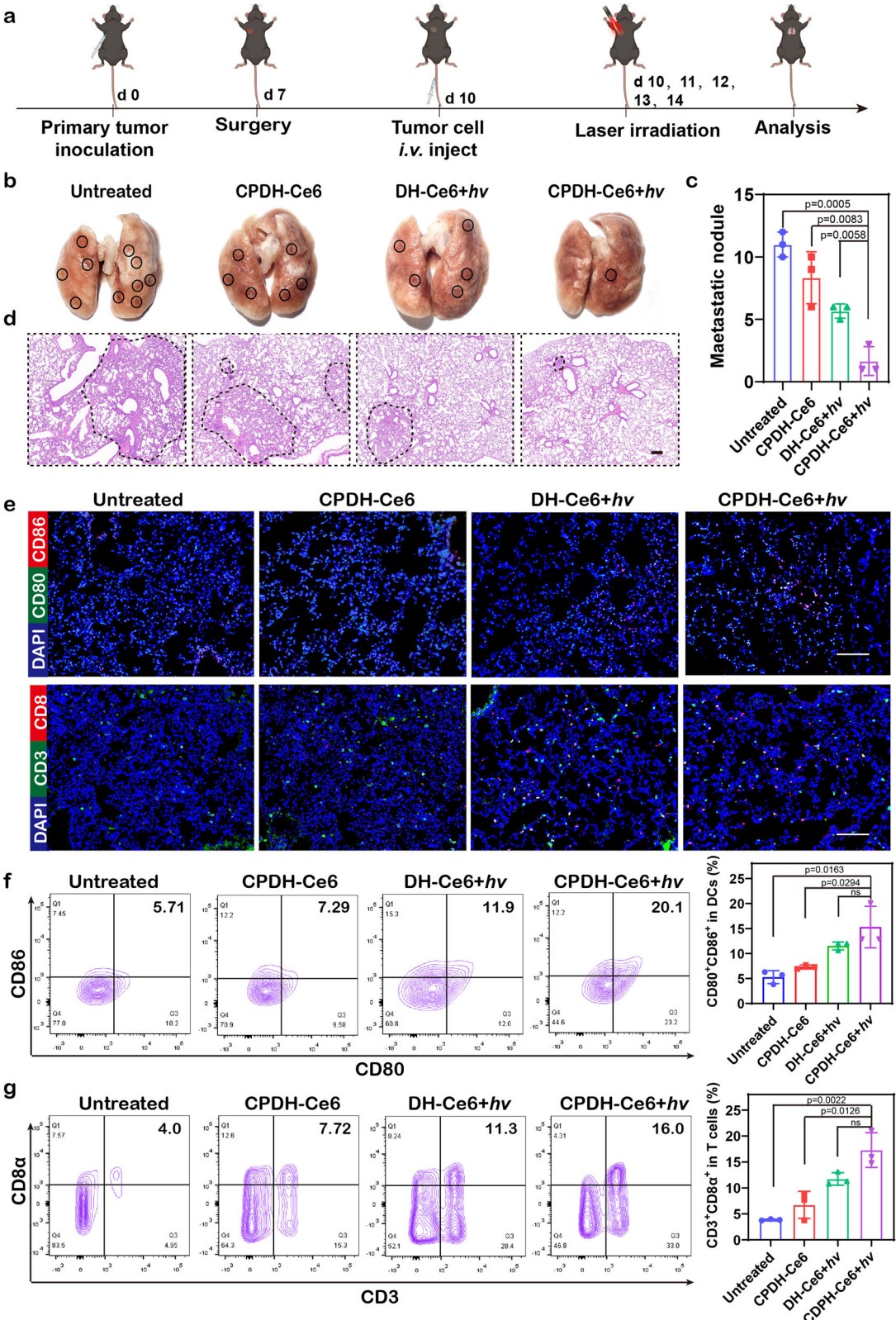

Red for 15 min and washed twice with PBS buffer, and CPDH-Ce6 was stained with SYBR Green II for 15 min. During cell enrichment, CPDH-Ce6 or CDH-Ce6 were combined with the cell suspension, respectively, and incubated at 37 °C for 1 h. The cell-captured hydrogels were washed with PBS buffer, then transferred to confocal dishes and imaged under a confocal microscope. The resulting images are processed using LAS AF Lite software (NIS ElementsAR ver. 5.02.01).

**Investigate the capture efficiency of CPDH-Ce6 on target cells**

B16F10 cells were cultured in an environment filled with 5% $CO_2$ and before use, cells are digested with trypsin and counted on a counting plate. In a classic cell enrichment procedure. The B16F10 cell suspension and different volumes of CPDH-Ce6 were incubated in a culture medium for 1 h. The resulting block hydrogel was washed 2 times with PBS buffer and then transferred to another tube containing RPMI 1640

**Fig. 7 | Postoperative metastasis inhibition in B16F10 tumor-bearing C57BL/6 mice by laser-irradiated CPDH-Ce6. a** Schematic diagram of the experimental design for the construction and treatment of lung metastasis animal models. **b** Representative photographs of lung tissues from mice in different groups. The black circles referred to the metastatic nodules. **c** The calculated lung metastasis nodules of mice after various treatments ($n = 3$ independent experiments). Data are analyzed by two-sided Student's $t$-test and shown as mean ± SD. **d** Representative H&E of lung sections from mice in different treatment groups. Scale bar: 200 μm. The dotted circles mark the damaged regions. The experiments were repeated

three times independently. **e** Immunofluorescence images showing DC (green: CD80[+], red: CD86[+]) and CTL (green: CD3[+] T cells, red: CD8[+] T cells) infiltration in the lungs of different groups. Scale bar: 100 μm. The experiments were repeated three times independently. **f** Representative flow cytometric analysis of DCs (CD80[+]CD86[+]) and **g** CTLs (CD3[+]CD8α[+]) in lymph nodes (tumor ipsilateral), and quantitative results of the different groups ($n = 3$ independent experiments). Data are analyzed by two-sided Student's $t$-test and shown as mean ± SD. Source data from (**c**, **f** and **g**) are provided as a Source Data file.

medium. To calculate capture efficiency, cells were first counted ($N_{total}$) in 10 μL of the target cell solution supernatant using a cell plate counter. Next, transfer enriched target cells for CPDH-Ce6 and count the cells in 10 μL of supernatant ($N_{remain}$). The capture rate is calculated as $(N_{total} − N_{remain})/N_{total} × 100\%$.

### In vitro detection of tumor cells
B16F10 cells were digested with trypsin and counted on a counting plate. Subsequently, the cells were placed into 100 μL of PBS to make cell suspension with different cell concentrations (0, 10, 20, 30, 40, 50, 100, 500, 1000, and 5000 cells/μL). The prepared CPDH@cAS was incubated with different densities of cells in the medium for 1 h at 37 °C. The gel was then removed, washed twice with PBS buffer, and transferred to a new tube. Fluorescence imaging was performed to observe the recovery of fluorescence signal from the detection module of CPDH@cAS. Finally, Bruker MR SE 3.2 software was used to analyze the experimental results.

### Cell cloning experiments
To evaluate the ability of CPDH-Ce6 to damage cells. The cells in the logarithmic growth phase were taken to prepare for cell suspension. Incubate B16F10 cell ($2 × 10^3$) suspension and CPDH-Ce6 (100 μL) in 10% RPMI 1640 complete medium at 37 °C for 1 h. The resulting block hydrogel was washed and then transferred to another tube containing RPMI 1640 medium. Then treated or not treated with 10 min of laser irradiation, CPDH was digested using 0.01 U of DNase I for 30 min and placed in a fresh medium until the cells formed colonies (~2 weeks). It is worth noting that the culture fluid. After the experiment, they were fixed in anhydrous methanol for 20 min. They were stained with 0.1% crystal violet for 15 min and photographed after washing. The number of cell clones in each sample was calculated using Image-J software (version 1.53a). Calculate the colony formation rate according to the following formula: (number of colonies in the experimental group/ number of colonies in the control group) × 100%.

### Analyze the in vivo stability of CPDH@cAS and CPDH@lAS
To construct a mouse model, B16F10 cells ($1 × 10^6$ cells) were injected percutaneously into the right side of female mice. When the tumor volume grew to 100–200 mm³ in size, we performed an operation to remove the tumor and close the wound. The mice were randomly divided into two groups and treated with CPDH@cAS and CPDH@lAS. Immediately after surgery, different probes were embedded in the hydrogel, and the wound was closed with absorbable sutures. Whole-body fluorescence imaging was subsequently performed at the indicated time points. After 48 h of embedding, mice were euthanized and tumors and major organs were harvested for ex vivo imaging. The warning signals were analyzed semi-quantitatively using Image-J analysis software (version 1.53a).

### Detecting early warning signs of tumor recurrence in vivo
After the mice were anesthetized, a portion of the tumor tissue was excised. CPDH@cAS, CDH@cAS, and cpDH@crAS were embedded in the excised site. Fluorescent images of live mice were collected at specific time points using an IVIS imaging system (Beckman coulter, USA). The early warning signals were analyzed by semi-quantitative

analysis. To observe the spatial distribution of cAS probes, CLSM imaging of tumor cryosections was performed, and image processing with LAS AF Lite software (NIS ElementsAR ver. 5.02.01).

### Analysis of tumor cells captured in vivo
B16F10 cells were digested and centrifuged to obtain a cell suspension, which was then incubated with Cell Mask™ Deep Red Plasma at 37 °C for 30 min to stain the tumor cells. DNA hydrogels, with or without PDL1 aptamer, were stained with SYBR Green II. The mice were anesthetized, and the CPDH and CDH were embedded in the surgical site where the tumor was previously removed and then sutured. The tumor cells, labeled with Cell Mask™ Deep Red Plasma, were injected in situ, avoiding the direct colocalization with hydrogel. After 2 h, the mice were frozen for 1 h, and frozen sections were prepared to observe the capture of tumor cells by the encapsulated DNA hydrogel using CLSM.

### Sensitivity analysis of early warning signals for detecting tumor recurrence in vivo
Before use, B16F10 cells were digested with trypsin and counted on a counting plate. The cell suspension was diluted into different amounts (0, 500, 1000, 5000, 10,000, and 20,000 cells) into new tubes. General anesthesia was performed by injecting sodium pentobarbital into the peritoneal cavity of mice, and CPDH@cAS hydrogels were surgically embedded in the axillae of mice. Different amounts of B16F10 cell suspension (50 μL) were injected into the postoperative site where the CPDH@cAS hydrogel had been embedded. Fluorescence imaging was performed to observe the recovery of fluorescence signal from the detection module of CPDH@cAS. Early warning signals were analyzed by the semi-quantitative fluorescence intensity of the ATP sensor.

### Stability analysis of CPDH-Ce6
To evaluate the stability of CPDH-Ce6 in vitro, we simulated the physiological environment with 10% FBS. The prepared CPDH, CPDH-Ce6, and CPDH-Ce6@cAS were incubated in 10% FBS and photographed daily to observe the stability of DNA hydrogels in vitro. To assess the in vivo stability of CPDH-Ce6, mice were anesthetized using sodium pentobarbital solution and most of the tumors were surgically removed. Subsequently, 100 μL of cDNA-Cy5 or CPDH-Ce6-Cy5 was implanted at the postoperative tumor resection site in mice, and the stability of the DNA hydrogel in mice was examined through daily live imaging of the small animals to record the fluorescence of Cy5.

### In vivo tumor models and treatment
All animal experiments were performed in accordance with the ethical guidelines approved by the Henan Provincial Animal Center (HNAC2018-0167). Mice were observed daily during the in vivo experiments and were euthanized when they showed a 20% weight loss. In some cases, this limit was exceeded on the last day of measurements and the mice were euthanized immediately. The maximum diameter of the tumor volume of the mice did not exceed 15 mm, which was approved by the Henan Provincial Animal Experimentation Ethics Committee.

To study the therapeutic effects of CPDH-Ce6, $1 × 10^6$ Luc-B16F10 cells were injected subcutaneously into the right axilla of mice. Five

days later, a second tumor was injected subcutaneously into the left axilla of each mouse as a distant tumor ($1 \times 10^6$ Luc-B16F10). Three days later, the mice were randomly divided into four groups (five mice per group), and the primary tumors were removed to mimic the tiny tumors remaining after surgery. Briefly, mice were anesthetized with sodium pentobarbital (10 mg/mL) and the tumor was removed using sterile instruments. Immediately after surgery, DNA hydrogels of different formulations were embedded, including CPDH-Ce6, DH-Ce6+$hv$, and CPDH-Ce6+$hv$. The wound was closed with absorbable sutures. After hydrogel embedding for 1 h, only the right axillary tumor was irradiated with the laser and not the left axillary tumor.

Tumors were visualized using an in vivo bioluminescence imaging system. After the mice were anesthetized, D-luciferin in DPBS (15 mg mL$^{-1}$) was injected intraperitoneally into each mouse at a dose of 10 μL g$^{-1}$. The mice were imaged 10 min later using the IVIS intravital imaging system. Quantify the region of interest as mean radiance (photon s$^{-1}$ cm$^{-2}$ sr$^{-1}$) using Living Image software (Bruker MI SE 721). In addition, the mice's body weight, primary tumor volume, and distant tumor volume were recorded every two days. Mice were euthanized at the end of treatment, and tumor tissues and major organs of treated mice were fixed overnight. Dissected tumor tissue and major organs were processed into sections. Tumor sections and major organs were stained with H&E. Tumor sections were also stained with TUNEL.

### Analyzing tumor-infiltrating immune cells with flow cytometry
On day 4 after embedding the hydrogel, the recurrent tumor was removed and excised into pieces. Then the recurrent tumor tissue was mixed with 10 μL of enzyme R, 100 μL of enzyme D, 12.5 μL of enzyme A and 2.35 mL of RPMI 1640, tumor tissues were incubated with shaking at 37 °C for 40 min. The supernatant from digested tumor tissue was collected through a 70 μm filter, centrifuged for 5 min, and resuspended to gain a single-cell suspension. The cell suspensions were stained for T cells with PE/Cyanine7 anti-mouse CD45 antibody (1/50 dilution), FITC CD3 antibody (1/50 dilution), and APC CD8α antibody (1/50 dilution). PE/Cyanine7 CD45 (1/50 dilution), Brilliant Violet 421 CD11c (1/50 dilution), PE CD80 (1/50 dilution), and Brilliant Violet 650 CD86 antibodies (1/50 dilution) were used for DCs detection. After washing twice with PBS, the cells were resuspended in 100 μL PBS and incubated on ice for flow cytometry. Data Files were analyzed using Flow-Jo 10.4 software.

### Detecting serum cytokine levels by ELISA
On the fourth postoperative day, blood was taken from the mice's eyeballs and the serum was separated. The levels of cytokines IL-6, IFN-γ, and TNF-α in the serum were measured by ELISA kits.

### Immunofluorescence staining
Mice were euthanized on the 4th postoperative day and immunofluorescent staining of tumor tissues (Anti-mouse Foxp3 Antibody 1:100 dilution, Anti-mouse PDL1 Antibody 1:100 dilution) was performed according to the manufacturer's instructions. In addition, frozen tumor tissues were used to examine the ROS produced at the tumor site.

### Lung metastasis analysis
A lung metastasis model was established to evaluate the anti-metastatic effect of CPDH-Ce6. Briefly, B16F10 cells ($1 \times 10^7$) were injected subcutaneously in the right axilla. Mice were randomly divided into 6 groups and post-operative excision and encapsulation of hydrogel preparations were performed on day 7. B16F10 cells were injected in the tail vein on day 10 and the surgical site was irradiated daily with a 660 nm laser (0.2 Wcm$^{-2}$,10 min). Mice were dissected after day 17 and lungs were removed, washed in PBS, and immediately placed in paraformaldehyde fixative.

The fixed lung tissue was examined meticulously under a high-resolution stereomicroscope to identify metastatic nodules. Moreover, fixed tissue was embedded in paraffin and sections were prepared, which were subsequently subjected to a histological staining technique with hematoxylin and eosin. The metastases present in each section were carefully counted by using a precise light microscope. In addition, the surface area occupied by the metastatic lesions was precisely quantified using Image-J software (version 1.53a).

### Analysis of immunomodulation in metastasis models
To analyze DC cell maturation and T cell infiltration in lung metastasis sites, immunofluorescent staining (CD3/CD8 and CD80/CD86) was performed on mouse lung tissue at the end of the lung metastasis model treatment according to the manufacturer's instructions. To analyze the immune cells in the lymph nodes, lymph nodes adjacent to the tumor were collected to study the infiltration of mature DCs and T cells in the tumor-draining lymph nodes. Analysis was performed by flow cytometry. All data were analyzed using Flow-Jo 10.4 software.

### Statistical analysis
Statistical analyses were performed using GraphPad Prism 8.0.2. All experiments were replicated independently at least three times. Unless otherwise stated, data are presented as mean ± SD of $n \geq 3$ independent biological replicates. Statistical significance between pairs of data was determined using unpaired two-sided Student's $t$-tests for continuous variables and one-way ANOVA with Tukey's correction and 95% confidence intervals for categorical variables. For all tests, n.s. meant not significant, $*P < 0.05$, $**P < 0.01$, $***P < 0.001$ and $****P < 0.0001$.

### Reporting summary
Further information on research design is available in the Nature Portfolio Reporting Summary linked to this article.

## Data availability
All data are available within the Article, Supplementary Information or Source Data file. Source data are provided with this paper.

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

## Acknowledgements

The authors gratefully acknowledge financial support from the National Natural Science Foundation of China (No. 22122409, U2004197); Henan Province Fund for Cultivating Advantageous Disciplines (No. 222301420019); Programs for Science & Technology Innovation Talents in Universities of Henan Province (No. 21HASTIT043). All the animal experiments were performed in accordance with the guidelines of the Regional Ethics Committee for Animal Experiments and Zhengzhou University Institutional Animal Care and Use Committee. The authors thank Modern Analysis and Computing Center of Zhengzhou University for technical assistance.

## Author contributions

K.X.Z., J.J.S., and D.Y.W. conceived and coordinated the study. Z.Z.Z., H.W.S., and J.J.L. supervised the project. D.Y.W. and J.W.L. performed the experiments. J.D. and H.Y. participated in performing experiments and discussing the results; D.Y.W. analyzed the data and wrote the paper. D.Y.W. and J.W.L. contributed equally to this work. All authors discussed the results and have given approval to the final version of the paper.

## Competing interests

The authors declare no competing interests.
