## [Peer Review File · Nature Communications]

Enrichment and sensing tumor cells by embedded immunomodulatory DNA hydrogel to inhibit postoperative tumor recurrenceREVIEWER COMMENTS

Reviewer #1 (Remarks to the Author):

In this manuscript, the authors developed a postoperative embedded DNA hydrogel for local enrichment of recurrent tumor cells to enable monitoring of tumor recurrence and timely photodynamic immunotherapy. This DNA hydrogel can enrich tumor cells in situ using PDL1 aptamers, triggering the loaded ATP sensor to provide monitoring signals for timely administration of adequate photodynamic immunotherapy. In addition, this in situ tumor cells enrichment strategy can be extended with other capture modules, detection modules, and treatment modules, showing great clinical translational potential. This is an interesting study and its conclusions are supported by the data. I recommend publishing it in Nature Communications after solving the following concerns.

Major issues

1. The in vivo PDL1 binding property of DNA hydrogel enables enrichment of tumor cells and blocking of immune checkpoints, which is the key design in this study. Why the author chose to use PDL1 aptamer, but not the widely used anti-PDL1 antibody, should be reasoned with more details.
2. In the process of synthesizing DNA hydrogel, the authors emphasized their study on adding photosensitizer (Ce6) at different time points, but didn't explain with adequate discussion. Does the addition of Ce6 at different time points significantly affect the formation of DNA hydrogel?
3. In Figure 2m, the Sanger sequencing results can only prove the presence of PDL1 aptamer and CpG sequences. The authors need to design a new experiment to assess the amount of intact functional sequence that can be released from DNA hydrogels under different conditions.
4. According to Figure 3b, it seems the tumor cells can only be captured on the surface of DNA hydrogel, would this affect tumor monitoring and treatment efficiency?
5. In this work, the authors performed in vitro test to evaluate the sensitivity of DNA hydrogel for sensing tumor cells. More detailed experiment process should be provided. Besides, the in vivo tumor cell detection sensitivity of DNA hydrogel should also be tested.
6. In Figure S12a, the expression of PDL1 protein in tumor tissues was significantly increased after the treatment. Please provide an explanation for this phenomenon.

Minor issues

1. Note the unity of writing forms involving free Ce6 and Ce6-cDNA in the manuscript.
2. Please pay attention to the use of singular and plural forms. E.g: In line 106, "was" changed to "were"; In line 175, "are" changed to "is", etc.
3. Please pay attention to the spelling of the word. E.g: In line 52, "AS" changed to "As"; In line 100, "hybridised" changed to "hybridized", etc.

Reviewer #2 (Remarks to the Author):

The manuscript aims to develop a system for detection of tumor recurrence, capture relapsed cells, and then provide for tumor cell killing by photodynamic therapy. The system is provocative, and the manuscript details several interesting results. The comments below details both areas of enthusiasm and concern with the results as well as the writing of the manuscript.

1. A major claim of the manuscript is that the system can detect recurrence. However, their model is an incomplete resection, which does not really recapitulate recurrence. Surgeons may miss a few cells, but are unlikely to miss 10% of the tumor. Similarly, the authors implant tumor cells at a second site and refer to that as a metastasis. Overall, the data seems to support the idea of locally treating a primary tumor rather than recurrence, and there are systemic effects that impact other tumor sites.
2. The detection of recurrence raises concerns about the fidelity of the system because the readings are only over 6 hours into an established tumor. For detecting recurrence, you would likely need persistence of the readout over much longer times. How stable are these materials in vivo – can they last for weeks to months? The results would be more compelling to see the ATP

levels start near zero (near complete resection) and then increase over days or weeks as the tumor recurred. The current study seems to only indicate that ATP can be measured at a partially resected tumor.

3. This results have some of the limitations of PDT – namely that they can slow tumor growth, but they cannot eliminate tumors. Fig 5.

4. The text states “PDL1 aptamer and CpG, together with TAA to induce immunomodulatory factors secretion”. What is the mechanism by which PDL1 would like to immune modulatory factor secretion. It is not clear that these studies really show this direct connection, more that the PDL1 gels did better at limiting tumor growth, but it is not clear that PDL1 directly contributed to the immune modulatory factors.

5. The manuscript claims to capture and enrich in situ relapsed tumor cells. No histology of the implant is shown to indicate the presence and enrichment of tumor cells in the implant. The in vitro assay for capture is not well described and I do not understand how it was performed. The assay does not seem to recapitulate the in vivo situation, an invasion assays into the gel would seem to be more appropriate. PDL1 is generally described as a signaling molecule not supporting adhesion, and thus this result is a bit confusing.

6. On page 15, the authors state “In contrast, due to the absence of tumor cell enrichment,”, but they do not show histology that would be necessary to conclude tumor cell enrichment or not.

7. The authors cite an inhibition rate of 88.1%, but it is not clear what that demonstrates. How is this calculated?

8. The paper has many acronyms and it is hard to keep track of them. ICD, and RCA were not defined, and others were defined late in the text.

9. The text states “Local presentation of DC cells to tumor antigens ...”, which was unclear. I assume DC cells are just dendritic cells. But the wording of presenting cells to tumor antigens is imprecise.

10. The text states “the proportion of Tregs (Foxp3+ T cells) was decreased, indicating an enhanced antigen-specific T cell response”. I am not sure how you can say it was antigen specific without sequencing the TCR.

11. The last sentence states “Collectively, these data suggest that hydrogels based on PDT combined with the photoresponsive release of DNA-based immunomodulators can induce a robust immune response and thus trigger the most effective tumor suppression.” Do the authors intend to say that the robust response triggers tumor suppression, it sounds like they are indicating a suppressive immune response at the tumor.

12. Authors write ... “suggesting that PDL1 aptamers and CpG play an essential role in the treatment.” The absence of PDL1 leads to growth suppression of 50-70%, so it is not clear that PDL1 is essential.

13. The manuscript indicates fluorescence with crAS, yet crAS nucleotides had been mutated. This reviewer is not understanding something.

14. The discussion section has no references. Are there no other early detection systems, aptamer systems, PDT approaches for detection or clearance, PDL1 signaling mechanisms, ... that are worthy of mentioning to provide context for the work that was performed in the manuscript?

Reviewer #3 (Remarks to the Author):

In this study, the authors describe and characterize a novel DNA-based hydrogel that captures tumor cells, identifies ATP signals produced by aggregated tumor cells, and contains PD-L1 aptamers. The stated application of this technology is to be able to detect residual cancer cells, capture them, and improve immunogenicity at the site through photodynamic therapy. This is an interesting approach towards a serious problem of minimal residual disease following surgical resection of a tumor. Overall interesting comment but needs additional experiments to back up claims the authors make and to demonstrate more proof-of-concept for clinical application.

A few questions:

1) what is the mechanism of tumor capture from PD-L1? Especially using the B16F10 model, which is a more immunogenic melanoma cell line.

2) Fig. 3E does not seem to have a wide dynamic range of signal between 10-50 cells/uL and statistical analysis of 3F would be more convincing. what if the cell density was increased 10 or

100 fold?

3) control experiment needed for Fig. 3G with laser irradiation of tumor cells in general - otherwise hard to know if anti-proliferative effect is solely driven by hydrogels.

4) At face value, the ATP detection with their system is impressive in Figure 4. However, the hydrogels were embedded after a partial resection where the MRD would be high. It is hard to imagine a scenario where the MRD burden would be so high after a partial resection, and therefore the design of this experiment limits its real-world application. Would like to see how sensitive the hydrogel is at lower concentrations of residual cells (i.e. full resection) and whether or not it can detect them.

5) The text connection with "And there was no non-specific signal in the major organs (heart, liver, spleen, lung, and kidney)

in all three groups, demonstrating that the hydrogel system has no significant toxic side effects in vivo (Figure 4e)." is not reflected in the figure.

6) The metastasis model used in Fig. 5 is not a natural seeding model, nor is it via accepted routes of metastasis for melanoma. It would be better to do this experiment in a spontaneous model of metastasis or other more accurate models of melanoma metastasis.

7) Would be interesting to see if immunomodulatory effect holds for TILs at the metastatic site as well, for Figure 6 experiments.

8) In Figure 6F, what type of statistical test was done and were other groups compared?

Minor:

- make abbreviations more clear throughout paper, (e.g. CDH is without the PD-L1 aptamer, not clearly explained in text)

- there are some grammatical errors in the text which could use more proofreading

- Please soften the language around treatment effect. For example, the hydrogel does not inhibit tumor recurrence and metastasis, but only abrogates it compared to untreated controls.

- the description of CD4+CD8+ T cells make it seem like the authors are looking for co-localized cells, but actually describing separate T cells

- language around claims throughout the text needs to be softened, such as "most effective tumor suppression", most effective compared to what?

Response to reviewers' comments

Reviewer #1

Comments:

In this manuscript, the authors developed a postoperative embedded DNA hydrogel for local enrichment of recurrent tumor cells to enable monitoring of tumor recurrence and timely photodynamic immunotherapy. This DNA hydrogel can enrich tumor cells in situ using PDL1 aptamers, triggering the loaded ATP sensor to provide monitoring signals for timely administration of adequate photodynamic immunotherapy. In addition, this in situ tumor cells enrichment strategy can be extended with other capture modules, detection modules, and treatment modules, showing great clinical translational potential. This is an interesting study and its conclusions are supported by the data. I recommend publishing it in Nature Communications after solving the following concerns.

Response:

Thanks for your appreciation for our work with instructive suggestions. Based on your suggestions, we revised the manuscript one by one as follows and marked **in blue**:

Question 1: *The in vivo PDL1 binding property of DNA hydrogel enables enrichment of tumor cells and blocking of immune checkpoints, which is the key design in this study. Why the author chose to use PDL1 aptamer, but not the widely used anti-PDL1 antibody, should be reasoned with more details.*

Response: Thank you for your valuable feedback and the opportunity to clarify our design choices. We chose to use the PDL1 aptamer rather than the anti-PDL1 antibody for several reasons. First, nucleic acid aptamers offer several advantages over antibodies, including ease of modification, low immunogenicity, and the ability to be synthesized on a large scale. [Angew Chem Int Ed Engl, e202214750 (2022)] [Science 334, 1716-1719 (2011)] [Nat Rev Drug Discov 16, 181-202 (2017)] Additionally, the programmable nature of nucleic acid aptamers allows them to be encoded in DNA hydrogel systems, which facilitates *in situ* capture and enrichment of tumor cells expressing the target protein. These features make the PDL1 aptamer a more suitable choice for our design, which aims to enrich tumor cells and block immune checkpoints *in vivo*.

We have included a more detailed description of these considerations in the Results and Discussion section of our manuscript to address your comment: **Nucleic acid aptamers offer several advantages over antibodies, including large-scale synthesis, ease of modification, and low immunogenicity.** [Angew Chem Int Ed Engl, e202214750 (2022)] [Science 334, 1716-1719

(2011)] [Nat Rev Drug Discov 16, 181-202 (2017)] Importantly, the programmable nature of nucleic acid aptamers enables the PDL1 aptamer to be encoded in DNA hydrogel systems. This allows the DNA hydrogel to specifically bind the PDL1 protein overexpressed on the surface of tumor cells, facilitating *in situ* capture and enrichment of tumor cells. As a result of these features, we selected the PDL1 aptamer as the cell enrichment module for our study.

Question 2: *In the process of synthesizing DNA hydrogel, the authors emphasized their study on adding photosensitizer (Ce6) at different time points, but didn't explain with adequate discussion. Does the addition of Ce6 at different time points significantly affect the formation of DNA hydrogel?*

Response: Thanks for your constructive suggestion. In order to thoroughly investigate the impact of adding Ce6 at different time points on DNA hydrogel formation, we conducted an experiment where we analyzed the effect of adding Ce6 at various stages of the RCA reaction. Specifically, we tested the effect of adding Ce6 at six different time points (0 h, 1 h, 2 h, 4 h, 6 h, and 8 h) on the ability of CPDH to load Ce6. Firstly, the fluorescence values of Ce6-cDNA were measured by a microplate reader. Next, Ce6-cDNA was added to 100 μ L of CPDH solution at different time points of the RCA reaction. At the end of the reaction, centrifugation was performed and the supernatant from each group was added to a 96-well plate. The fluorescence value of Ce6-cDNA in the supernatant of each group was measured. The fluorescence difference was used to calculate the loading of Ce6-cDNA by CPDH. The results of this experiment are presented in **Figure R1**.

Figure R1. Encapsulation rate of Ce6 added during RCA reaction (n = 3).

The results showed that introducing the photosensitizer Ce6 at different stages during the DNA hydrogel synthesis process had a noteworthy influence on its loading capacity. The optimal

point to incorporate Ce6 during the RCA reaction was after 2 h to achieve the maximum loading efficiency.

To explore the effect of Ce6 addition timing on hydrogel formation, we also examined the morphology of hydrogel formation through visual photographs and cryo-scanning electron microscopy (**Figure R2**). Specifically, CPDH-Ce6 was obtained by adding Ce6 at different time points of the RCA reaction. The reaction was carried out at 37°C and terminated at 80°C for 20 min to inactivate the polymerase and other active substances. The prepared DNA hydrogels were stained with SYBR Green II for 15 min and washed twice with PBS at the end of staining, then placed under UV light to observe the morphology of the hydrogels formed by adding Ce6 at different time points. In addition, the morphology of the DNA hydrogels formed by adding Ce6 at different time points was observed by cryo-scanning electron microscopy.

Figure R2. The DNA hydrogels synthesized by adding Ce6 at different times of the RCA reaction were visualized in pictures a) and cryo-SEM b). Scale bar: 50 μm.

The addition of Ce6-cDNA at 0 h and 1 h of the RCA reaction had a significant impact on hydrogel formation. However, adding Ce6-cDNA at different time points after 2 h (2 h, 4 h, 6 h, 8 h) did not significantly affect the formation of DNA hydrogels. We propose that this

phenomenon is likely due to the interference of Ce6-cDNA addition at the beginning of the RCA reaction with the partially complementary long single-stranded DNA template for nucleic acid hybridization. Therefore, we decided to add Ce6-cDNA after 2 h of RCA reaction for DNA hydrogel synthesis.

We revised as followed:

“Additionally, we observed that adding Ce6-cDNA at 0 h and 1 h significantly impacted the formation of hydrogels, whereas adding Ce6-cDNA at different time points after 2 h (2 h, 4 h, 6 h, 8 h) did not significantly interfere with the formation of DNA hydrogels. Therefore, we selected the time point of 2 h in the RCA reaction to add Ce6-cDNA for the synthesis of DNA hydrogels (Figure S4).” was added to the Results and Discussion section as marked in blue.

“*The effect of the addition of Ce6-cDNA on hydrogel formation during the RCA reaction: specifically, we tested the effect of adding Ce6-cDNA at six different time points (0 h, 1 h, 2 h, 4 h, 6 h, and 8 h) on the ability of CPDH to load Ce6-cDNA. Firstly, the fluorescence values of Ce6-cDNA were measured by a microplate reader. Next, Ce6-cDNA was added to 100 μ L of CPDH solution at different time points of the RCA reaction. At the end of the reaction, centrifugation was performed and the supernatant from each group was added to a 96-well plate. The fluorescence value of Ce6-cDNA in the supernatant of each group was measured. The fluorescence difference was used to calculate the loading of Ce6-cDNA by CPDH.*

Next, the effect of adding Ce6-cDNA at different times during the RCA reaction on the morphology of the hydrogels was examined using visual images and cryo-scanning electron microscopy. Specifically, CPDH-Ce6 was obtained by adding Ce6-cDNA at different time points of the RCA reaction. The reaction was carried out at 37°C and terminated at 80°C for 20 min to inactivate the polymerase and other active substances. The prepared DNA hydrogels were stained with SYBR Green II for 15 min and washed twice with PBS at the end of staining, then placed under UV light to observe the morphology of the hydrogels formed by adding Ce6-cDNA at different time points. In addition, the morphology of the DNA hydrogels formed by adding Ce6-cDNA at different time points was observed by cryo-scanning electron microscopy.” was added to the Methods section as marked in blue.

Question 3: *In Figure 2m, the Sanger sequencing results can only prove the presence of PDL1 aptamer and CpG sequences. The authors need to design a new experiment to assess the amount of intact functional sequence that can be released from DNA hydrogels under different conditions.*

Response: Thanks for your constructive suggestion. To examine the release of intact functional PDL1 aptamer and CpG sequences from DNA hydrogels, we utilized a fluorescence-based molecular beacon assay to analyze the functional PDL1 aptamer and CpG sequences exposed to solution before and after light irradiation (**Figure R3**). Specifically, CPDH-Ce6 (with or without 10 min of laser irradiation) was centrifuged. The supernatant was taken and incubated with the molecular beacon probe for 30 min at 37°C. Finally, it was added to a 96-well plate. Fluorescence recovery of the molecular beacon probe was measured by microplate reader to assess the integrity of the CpG and PDL1 aptamers.

Figure R3. Fluorescence-based molecular beacon test to estimate the integrity and accessibility of functional AptPDL1 and CpG released by CPDH-Ce6 (with or without 10 min laser irradiation). **** $P < 0.0001$ determined by Student's t-test.

The results showed that a significant increase in the quantity of PDL1 aptamer and CpG sequences exposed in solution after irradiation, with a 3.2- and 5.2-fold increase, respectively, compared to the control group. Our findings demonstrate that the light-responsive DNA hydrogels have the ability to release fully functional PDL1 aptamers and CpG sequences, which holds potential for the development of innovative drug delivery systems in the field of immunotherapy.

Furthermore, to evaluate the proportion of intact and functional PDL1 aptamer and CpG sequences that were released from DNA hydrogels, molecular beacons were incubated with laser-irradiated CPDH-Ce6 hydrogels synthesized by 10 μM primer/circle hybridization at 37°C for 30 min. We calculated the release rate using long single-stranded DNA synthesized by 10 μM primer/circle hybridization as a control (**Figure R4**).

Figure R4. Characterization of the release rates of functional PDL1 aptamers and CpG sequences released from laser irradiated CPDH-Ce6.

The release rates of the functional PDL1 aptamer and CpG sequences were found to be 70.1% and 63.9%, respectively, indicating that irradiation had caused the majority of functional sequences to be released from CPDH-Ce6.

We revised as followed:

“Furthermore, to evaluate the proportion of intact and functional PDL1 aptamer and CpG sequences that were released from DNA hydrogels, molecular beacons were incubated with laser-irradiated CPDH-Ce6 hydrogels synthesized by 10 μ M primer/circle hybridization at 37°C for 30 min. We calculated the release rate using long single-stranded DNA synthesized by 10 μ M primer/circle hybridization as a control. The release rates for the functional PDL1 aptamer and CpG sequences were identified as 70.1% and 63.9%, respectively. This indicates that irradiation induced the release of the majority of the functional sequences from CPDH-Ce6” was added to the Results and Discussion section as marked in blue.

Question 4: *According to Figure 3b, it seems the tumor cells can only be captured on the surface of DNA hydrogel, would this affect tumor monitoring and treatment efficiency?*

Response: Thanks for your comments. Our study involves a modification of the ATP circular probe on the surface of the DNA hydrogel, which serves as the detection module. By capturing the majority of tumor cells on the surface of the hydrogel, we bring these cells closer to the probe, resulting in a more timely and efficient detection signal. Therefore, the tumor cells captured on the hydrogel surface provide several benefits for tumor recurrence detection and photodynamic immunotherapy. Overall, our approach represents a promising strategy for improving the efficacy of tumor recurrence detection and treatment. By optimizing the capture of tumor cells

on the hydrogel surface and utilizing our modified ATP circular probe, we believe that our approach has the potential to significantly impact the field of cancer diagnosis and therapy.

Question 5: *In this work, the authors performed in vitro test to evaluate the sensitivity of DNA hydrogel for sensing tumor cells. More detailed experiment process should be provided. Besides, the in vivo tumor cell detection sensitivity of DNA hydrogel should also be tested.*

Response 5-1: Thank you for the insightful comment. In response to your feedback, we have added a detailed experimental procedure for assessing the sensitivity of our DNA hydrogels to tumor cell sensing *in vitro*. Specifically, we placed CPDH@cAS hydrogels into tubes and added 100 μ L of PBS to make cell suspension with different cell concentration (0, 10, 20, 30, 40, 50, 100, 500, 1000, and 5000 cells/ μ L) of tumor cells. After incubation at 37°C for 1 h, the hydrogels were gently washed twice with PBS and subsequently imaged using IVIS Spectrum. We believe that this approach tested the sensitivity of our hydrogels to tumor cell sensing and provides an assessment of their performance *in vitro*.

Response 5-2: To fully characterize the sensitivity of our DNA hydrogels for detecting relapsed tumor cells *in vivo*, we constructed a new model for detecting residual cells by injecting different numbers of tumor cells at the site of complete resection surgery. The B16F10 cell suspension was diluted into different amounts (0, 500, 1000, 5000, 10,000 and 20,000 cells) into new tubes. General anesthesia was performed by injecting sodium pentobarbital into the peritoneal cavity of mice, and CPDH@cAS hydrogels were surgically embedded in the axillae of mice. Different amounts of B16F10 cell suspension (50 μ L) were injected into the postoperative site where the CPDH@cAS hydrogel had been embedded. Fluorescence imaging was subsequently performed to observe the recovery of the fluorescence signal from the detection module of CPDH@cAS (**Figure R5**).

Figure R5. a) Schematic diagram of detecting tumor recurrence *in vivo*. b) Fluorescence imaging of different tumor cells injected after *in vivo* encapsulation of CPDH@cAS. c) Relative fluorescence quantification of corresponding fluorescence images in b). d) CPDH@cAS detected the signal intensity of different densities of tumor cells at 1 h.

The experimental results showed that the ATP sensor exhibited high sensitivity in detecting tumor cells, as indicated by the increasing fluorescence signal with higher cell counts. This

suggests that the ATP sensor has the potential to detect even small amounts of tumor cells (500 cells). Additionally, the ATP sensor was able to respond rapidly and reached its highest value at 1 h. The rapid response of the sensor to tumor cells is an important feature that could potentially enable monitoring the progression of cancer and guide treatment decisions. We believe that this approach effectively characterizes the sensitivity of our DNA hydrogels for detecting tumor cells *in vivo* and provides a more comprehensive evaluation of the DNA hydrogel's performance.

We revised as followed:

“The CDH and CPDH hydrogels were placed into tubes, followed by the addition of 100 μ L cell culture medium containing variable numbers of tumor cells, then incubated at 37°C for 1 h, washed gently twice with PBS, and subsequently imaged by IVIS Spectrum. The results showed that PDL1 aptamer played a crucial role in the capture and enrichment of tumor cells in DNA hydrogel (Figure 3b, 3c).” was added to the Results and Discussion section as marked in blue.

“Furthermore, we conducted an analysis on the sensitivity of CPDH@cAS for *in vivo* tumor cell detection by administering varying numbers of tumor cells (0, 500, 1000, 5000, 10000, and 20000 cells count) at the site of fully resected surgery (Figure S12). The obtained experimental results demonstrated a positive correlation between the fluorescence generated by the ATP sensor and the number of tumor cells, highlighting the high sensitivity of the sensor in detecting tumor recurrence. In addition, our findings revealed that the ATP sensor is capable of responding rapidly to the relapse of tumor cells and reaches its maximum value at 1 h, emphasizing its timely nature.” was added to the Results and Discussion section as marked in blue.

“*In vitro* detection of tumor cells: B16F10 cells were digested with trypsin and counted on a counting plate. Subsequently, the cells were placed into 100 μ L of PBS to make cell suspension with different cell concentrations (0, 10, 20, 30, 40, 50, 100, 500, 1000, and 5000 cells/ μ L). The prepared CPDH@cAS was incubated with different densities of cells in the medium for 1 h at 37°C. The gel was subsequently removed, washed twice with PBS buffer, and transferred to a new tube. Fluorescence imaging was performed to observe the recovery of fluorescence signal from the detection module of CPDH@cAS. Finally, Bruker MR SE software was used to analyze the experimental results.” was added to the Methods section as marked in blue.

“*Sensitivity analysis of early warning signals for detecting tumor recurrence in vivo*: Before using, B16F10 cells were digested with trypsin and counted on a counting plate. The cell suspension was diluted into different amounts (0, 500, 1000, 5000, 10,000 and 20,000 cells) into new tubes. General anesthesia was performed by injecting sodium pentobarbital into the

peritoneal cavity of mice, and CPDH@cAS hydrogels were surgically embedded in the axillae of mice. Different amounts of B16F10 cell suspension (50 μ L) were injected into the postoperative site where the CPDH@cAS hydrogel had been embedded. Early warning signals were imaged using IVIS Spectrum and analyzed by semi-quantitative fluorescence intensity of the ATP sensor.” was added to the Methods section as marked in blue.

Question 6: *In Figure S12a, the expression of PDL1 protein in tumor tissues was significantly increased after the treatment. Please provide an explanation for this phenomenon.*

Response: Regarding the increased expression of PDL1 in tumor tissue after treatment, we agree that there are multiple possible mechanisms that could contribute to this phenomenon. One possible explanation is that the treatment may cause damage to tumor cells, triggering an immune response. As part of this immune response, immune cells such as T cells and macrophages can release cytokines that stimulate the upregulation of PDL1 expression in tumor cells. This immune-mediated upregulation of PDL1 has been observed in other studies and may represent a key mechanism in cancer immunotherapy. [Cancer Lett 476, 170-182 (2020)] [Nat Rev Immunol 20, 209-215 (2020)]

In addition, the increase in PDL1 expression may also be the result of a feedback mechanism activated in response to treatment. This feedback mechanism can be triggered by the activation of various immune cells that can release cytokines that stimulate the upregulation of PDL1 expression in tumor cells. This mechanism may contribute to the sustained expression of PDL1 over time, even after treatment has ended. [J Am Chem Soc 143, 8391-8401 (2021)]

We revised as followed:

“The PDL1 protein is typically expressed at low levels in healthy tissues but becomes upregulated in tumor microenvironments, rendering it a promising target for immune checkpoint inhibition therapies. Upon binding to PDL1 on the tumor cell surface, the PDL1 aptamer may interfere with PDL1 protein function, leading to compensatory overexpression of PDL1 by the tumor cells and ultimately reinforcing the anti-tumor immune therapy. [Cancer Lett 476, 170-182 (2020)] [Nat Rev Immunol 20, 209-215 (2020)] [J Am Chem Soc 143, 8391-8401 (2021)]” was added to the Results and Discussion section as marked in blue.

Question 7: *Note the unity of writing forms involving free Ce6 and Ce6-cDNA in the manuscript.*

Response: Thank you for bringing this to our attention. We apologize for any confusion this may have caused. We will carefully review the manuscript to ensure consistency in the writing forms used for free Ce6 and Ce6-cDNA throughout the revised manuscript.

Question 8: *Please pay attention to the use of singular and plural forms. E.g: In line 106, “was” changed to “were”; In line 175, “are” changed to “is”, etc.*

Response: We appreciate your feedback and have carefully reviewed the manuscript and made the necessary changes to ensure consistency in the use of singular and plural forms. Specifically, we have changed "was" to "were" in line 106, and changed "are" to "is" in line 175, and have ensured that the singular or plural forms are used appropriately throughout the manuscript. We have also reviewed the figures and tables to ensure consistency with the use of singular and plural forms, as appropriate.

Question 9: *Please pay attention to the spelling of the word. E.g: In line 52, “AS” changed to “As”; In line 100, “hybridised” changed to “hybridized”, etc.*

Response: We are very grateful for your careful review. We have double-checked all the spelling issues in the whole manuscript to make sure there are no other mistakes.

Reviewer #2

Comments:

The manuscript aims to develop a system for detection of tumor recurrence, capture relapsed cells, and then provide for tumor cell killing by photodynamic therapy. The system is provocative, and the manuscript details several interesting results. The comments below details both areas of enthusiasm and concern with the results as well as the writing of the manuscript.

Response:

Thanks for your appreciation for our work with instructive suggestions. Based on your suggestions, we revised the manuscript one by one as follows and marked in blue:

Question 1: *A major claim of the manuscript is that the system can detect recurrence. However, their model is an incomplete resection, which does not really recapitulate recurrence. Surgeons may miss a few cells, but are unlikely to miss 10% of the tumor. Similarly, the authors implant tumor cells at a second site and refer to that as a metastasis. Overall, the data seems to support the idea of locally treating a primary tumor rather than recurrence, and there are systemic effects that impact other tumor sites.*

Response 1-1: Thank you for your insightful comments. We appreciate your comments on our study and agree with your opinion. Although in many research efforts, post-operative recurrence models are constructed by incomplete removal of the tumor, [Nat Commun 13, 4553 (2022)]

[Nature Nanotech 14, 89–97 (2019)] [Nat Biomed Eng 1, 0011 (2017)] we agree that it can simulate postoperative tumor recurrence to some extent, but could not fully replicate the postoperative tumor situation. To overcome this limitation, we performed new experiments that involved injecting different amounts of tumor cells (0, 500, 1000, 5000, 10,000, and 20,000 cell counts) at the fully resected surgical site. With this modified model, we aimed to better simulate post-operative recurrence and assess the sensitivity of DNA hydrogels for *in vivo* tumor cell detection. Specifically, B16F10 cells were digested with trypsin and counted on a counting plate. The cell suspension was diluted into different amounts (0, 500, 1000, 5000, 10,000 and 20,000 cells) into new tubes. General anesthesia was performed by injecting sodium pentobarbital into the peritoneal cavity of mice, and CPDH@cAS hydrogels were surgically embedded in the axillae of mice. Different amounts of B16F10 cell suspension (50 μ L) were injected into the postoperative site where the CPDH@cAS hydrogel had been embedded. Fluorescence imaging was performed to observe the recovery of fluorescence signal from the detection module of CPDH@cAS (**Figure R6**).

Figure R6. a) Schematic diagram of detecting tumor recurrence *in vivo*. b) Fluorescence imaging of different tumor cells injected after *in vivo* encapsulation of CPDH@cAS. c) Relative fluorescence quantification of corresponding fluorescence images in b). d) CPDH@cAS detected the signal intensity of different densities of tumor cells at 1 h.

The experimental results showed that the ATP sensor exhibited high sensitivity in detecting tumor cells, as indicated by the increasing fluorescence signal with higher cell counts. This suggests that the ATP sensor has the potential to detect even small amounts of tumor cells (500 cells). Additionally, the ATP sensor was able to respond rapidly and reached its highest value at 1 h. The rapid response of the sensor to tumor cells is an important feature that could potentially enable monitoring the progression of cancer and guide treatment decisions. We believe that this approach effectively characterizes the sensitivity of our DNA hydrogels for detecting tumor cells *in vivo* and provides a more comprehensive evaluation of the DNA hydrogel's performance.

Response 1-2: Thank you for your insightful comments. Although some research referred to the secondary location as metastasis [Adv. Mater. 2204765, (2022)] [ACS Nano 13, 12148–12161 (2019)], we acknowledge that the experimental design does not fully replicate the process of metastasis. To evaluate the inhibitory potential of photodynamic immunotherapy employing DNA hydrogels on tumor metastasis, we have established and treated a standard lung metastasis model of B16F10 tumors (**Figure R7a**). Briefly, we subcutaneously inoculated B16F10 cells (1×10^7) in the right axilla of mice. Mice were randomly divided into 6 groups and post-operative excision and encapsulation of hydrogel preparations were performed on day 7. B16F10 cells were intravenously injected on day 10 and the surgical site was irradiated daily with a 660 nm laser (0.2 Wcm^{-2} , 10 min). On day 17, mice were dissected and their lungs were removed for evaluation of lung metastases by photographs and histological analysis of H&E staining (**Figure R7b, Figure R7c**). In addition, to assess the recruitment of immune cells in the lung, we stained frozen lung sections with immunofluorescence, followed by confocal laser scanning microscopy (CLSM) imaging and semi-quantitative analysis (**Figure R7d, Figure R7e**).

Figure R7. Postoperative metastasis inhibition in B16F10 tumor-bearing C57BL/6 mice by laser-irradiated CPDH-Ce6. a) Schematic diagram of the experimental design for the construction and treatment of lung metastasis animal models. b) Representative photographs of lung tissues from mice in different groups. The black circles referred to the metastatic nodules. c) Representative H&E of lung sections from mice in different treatment groups. Scale bar: 200

μm . The dotted circles mark the damaged regions. d) Immunofluorescence images showing DC (red: CD80^+ , green: CD86^+) and CTL (green: CD3^+ T cells, red: CD8^+ T cells) infiltration in the lungs of different groups. Scale bar. $100 \mu\text{m}$. e) Analysis of fluorescence intensity of CD80^+ and CD86^+ DC cells, CD3^+ and CD8^+ T cells in lung sections ($n = 3$).

The experimental results showed that there were no obvious visible metastatic nodules in CPDH-Ce6+*hν*-treated mice, suggesting the CPDH-Ce6+*hν*-treated mice had a lower incidence of postoperative lung metastases compared to the control and other groups of mice. The proportion of CD3^+ T cells and CD8^+ T cells, CD80^+ DC cells and CD86^+ DC cells was significantly increased in the lung tissue of CPDH-Ce6+*hν*-treated mice, which was 1.7-fold, 6.9-fold, 2.1-fold, and 2.8-fold higher, respectively, than that of mice in the untreated group. These results suggest that this DNA hydrogel system can trigger intense photodynamic immunotherapy. The increased activation of immune cells may have contributed to the suppression of tumor recurrence and lung metastases.

We revised as followed:

“In addition, we characterized the sensitivity of CPDH@cAS for *in vivo* tumor cell detection by injecting different numbers of tumor cells at the fully resected surgical site (Figure S12). The experimental results showed that the ATP sensor generated more fluorescence as the cell count increased, indicating its high sensitivity in detecting tumor recurrence. Moreover, once the tumor cells relapsed, the ATP sensor was able to respond rapidly and reached its highest value at 1 h, indicating its timely nature.” was added to the Results and Discussion section as marked in blue.

“2.6 Evaluation of DNA hydrogel for suppression of lung metastases after surgical resection of primary tumors

Lung metastasis is another major clinical problem in malignant cancer patients after surgery. To evaluate the effect of DNA hydrogel treatment on lung metastases after surgical resection of primary tumors, we constructed and treated a standard B16F10 lung metastasis model (Figure 7a). Then, lung tissue was collected from each group of mice for analysis and comparison ($n = 3$). Lung metastases were assessed by photo graphs and histological analysis of H&E staining. There were no obvious visible metastatic nodules in CPDH-Ce6+*hν*-treated mice compared to control and other groups of mice, suggesting effective prevention of postoperative lung metastases (Figure 7b, Figure 7c).

To assess the recruitment of immune cells in the lung, we stained frozen lung sections by immunofluorescence followed by CLSM imaging. As shown in Figure 7d and Figure S18, the proportion of CD3⁺ T cells, CD8⁺ T cells, CD80⁺ DC cells, and CD86⁺ DC cells was significantly increased in the lung tissue of CPDH-Ce6+hv-treated mice, which was 1.7-fold, 6.9-fold, 2.1-fold, and 2.8-fold higher, respectively, compared to the untreated group of mice. These findings suggest that the treatment was effective in stimulating the immune response in the mice's lungs, as indicated by the increased presence of T cells and DC cells.” was added to the Results and Discussion section as marked in blue.

“*Sensitivity analysis of early warning signals for detecting tumor recurrence in vivo:* Before using, cells were digested with trypsin and counted on a counting plate. The B16F10 cell suspension was diluted into different amounts (0, 500, 1000, 5000, 10,000 and 20,000 cells) into new tubes. General anesthesia was performed by injecting sodium pentobarbital into the peritoneal cavity of mice, and CPDH@cAS hydrogels were surgically embedded in the axillae of mice. Different amounts of B16F10 cell suspension (50 μL) were injected into the postoperative site where the CPDH@cAS hydrogel had been embedded. Fluorescence imaging was performed to observe the recovery of fluorescence signal from the detection module of CPDH@cAS. Early warning signals were analyzed by semi-quantitative fluorescence intensity of the ATP sensor.” was added to the Methods section as marked in blue.

“*Lung metastasis analysis:* A lung metastasis model was established to evaluate the anti-metastatic effect of CPDH-Ce6. Briefly, B16F10 cells (1×10^7) were injected subcutaneously in the right axilla. Mice were randomly divided into 6 groups and post-operative excision and encapsulation of hydrogel preparations were performed on day 7. B16F10 cells were injected in the tail vein on day 10 and the surgical site was irradiated daily with a 660 nm laser (0.2 Wcm⁻², 10 min). Mice were dissected after day 17 and lungs were removed for visual photography and H&E staining.” was added to the Methods section as marked in blue.

“*Analysis of immunomodulation in metastasis models:* To analyze DC cell maturation and T cell infiltration in lung metastasis sites, immunofluorescent staining (CD3/CD8 and CD80/CD86) was performed on mouse lung tissue at the end of lung metastasis model treatment according to the manufacturer's instructions. All data were analyzed using Flow-Jo software.” was added to the Methods section as marked in blue.

Question 2: *The detection of recurrence raises concerns about the fidelity of the system because the readings are only over 6 hours into an established tumor. For detecting recurrence, you would likely need persistence of the readout over much longer times. How stable are these*

materials *in vivo* – can they last for weeks to months? The results would be more compelling to see the ATP levels start near zero (near complete resection) and then increase over days or weeks as the tumor recurred. The current study seems to only indicate that ATP can be measured at a partially resected tumor.

Response 2-1: Thank you for your thoughtful comments on our study. The stability of the hydrogel system is the basis for the detection and treatment of tumor recurrence in post-operative period. We added experiments to assess the stability of DNA hydrogels *in vitro* and *in vivo* (**Figure R8**). We firstly simulated a physiological environment *in vitro* with 10% FBS. The prepared CPDH, CPDH-Ce6 and CPDH-Ce6@cAS were placed in 10% FBS and photographed daily to observe the stability of the DNA hydrogels. We found that the DNA hydrogels could be stored morphologically stable for 10 days *in vitro*.

Figure R8. Morphological visualization of DNA hydrogels stored in 10% FBS for 14 days.

To evaluate the *in vivo* stability of CPDH-Ce6, mice were anesthetized with sodium pentobarbital solution (**Figure R9**). Most of the tumors were removed using surgery, with tiny remaining tumors to simulate tumor recurrence. We encapsulated fluorescent group Cy5-labelled Cy5-cDNA and CPDH-Ce6-Cy5 at the surgical site *in vivo*. The fluorescence of Cy5 was recorded by daily live imaging of the small animals to detect the stability of the DNA hydrogel in the mice.

Figure R9. In vivo imaging characterizing the *in vivo* retention of Cy5 fluorescence in Cy5-cDNA and CPDH-Ce6-Cy5 encapsulated at the surgical site in mice.

The fluorescence of Cy5-cDNA almost disappeared after about 1 day in mice, whereas the fluorescence intensity of the DNA hydrogels was still detectable after 6 days. The experimental results show that the DNA hydrogel can remain stable *in vitro* and *in vivo* for about one week, which can meet the detection needs of our research work. The long-term stability of DNA hydrogels may be further improved by combining them with polymers or transitioning to L-nucleic acid scaffolds. [Angew Chem Int Ed Engl, e202202520 (2022)] [Adv Drug Deliv Rev 168, 79-98 (2021)]

We revised as followed:

“The stability of the hydrogel system is the basis for its *in vivo* applications. We next assessed DNA hydrogel stability *in vitro* and *in vivo*. As shown in Figure S10a, we simulated a physiological environment *in vitro* with 10% FBS and found that the DNA hydrogels could be stored morphologically stable for 10 days. Then, we embedded Cy5-cDNA and CPDH-Ce6-Cy5 at the surgical site *in vivo*. As shown in Figure S10b, the fluorescence of Cy5-cDNA almost disappeared after about 1 day in mice, whereas the fluorescence intensity of DNA hydrogels was still detectable after 6 days, indicating that the DNA hydrogels can remain stable *in vitro* and *in vivo* for about 1 week. The long-term stability of DNA hydrogels may be further improved by combining them with polymers or transitioning to L-nucleic acid scaffolds.” was added to the Results and Discussion section as marked in blue.

“*Stability analysis of CPDH-Ce6:* To evaluate the stability of CPDH-Ce6 *in vitro*, we simulated the physiological environment with 10% FBS. The prepared CPDH, CPDH-Ce6 and CPDH-Ce6@cAS were incubated in 10% FBS and photographed daily to observe the stability

of DNA hydrogels *in vitro*. To assess the *in vivo* stability of CPDH-Ce6, mice were anesthetized using sodium pentobarbital solution and most of the tumors were surgically removed. Subsequently, 100 μ L of cDNA-Cy5 or CPDH-Ce6-Cy5 was implanted at the postoperative tumor resection site in mice, and the stability of the DNA hydrogel in mice was examined through daily live imaging of the small animals to record the fluorescence of Cy5.” was added to the Methods section as marked in blue.

Response 2-2: To improve the biological stability of the detection module *in vivo*, we designed a bicyclic ATP sensor (cAS) for the DNA hydrogel system, and studied the ability of ATP sensor to maintain signal readings within 48 h (**Figure R10**). Specifically, to construct a mouse model, B16F10 cells (1×10^6 cells) were injected percutaneously in the right side of female mice. When the tumor volume grew to 100 - 200 mm³ in size, we performed an operation to remove the tumor and close the wound. The mice were randomly divided into two groups and treated with CPDH@cAS and CPDH@IAS. Immediately after surgery, different probes were embedded in the hydrogel, and the wound was closed with absorbable sutures. Whole-body fluorescence imaging was subsequently performed at the indicated time points. After 48 h of embedding, mice were euthanized and tumors and major organs were harvested for *ex vivo* imaging. Semi-quantitative analysis of the warning signals was performed.

Figure R10. a) *In vivo* fluorescence images of mice with tumor recurrence after embedding different hydrogel sensors. b) Fluorescence images of isolated tumors and organs (heart, liver, spleen, lung, kidney) harvested 48 h after embedding. c) Cy5 fluorescence intensity of circular ATP sensor and linear ATP sensor in tumor region at different imaging times. Results are presented as means \pm standard deviation (SD) (n = 3). **P < 0.01 determined by Student's t-test.

The experimental results demonstrate that, following surgical implantation, the cAS group showed a fluorescence signal 1.6-fold higher than that of the lAS group after 48 h. This finding suggests that the cAS sensor is more stable *in vivo* than the lAS sensor, providing the possibility for long-term, stable detection of tumor recurrence.

Response 2-3: To better simulate the ATP levels in the early stages of tumor recurrence, we constructed a new model for detecting residual cells by injecting different numbers of tumor cells at the site of complete resection surgery. We utilized this model to assess the timely detection of ATP sensor (**Figure R11a**). Firstly, the B16F10 cell suspension was diluted into different amounts (0, 500, 1000, 5000, 10,000 and 20,000 cells) into new tubes. General anesthesia was performed by injecting sodium pentobarbital into the peritoneal cavity of mice, and CPDH@cAS hydrogels were surgically embedded in the axillae of mice. Different amounts of B16F10 cell suspension (50 μ L) were injected into the postoperative site where the CPDH@cAS hydrogel had been embedded. Fluorescence imaging was performed to observe the recovery of fluorescence signal from the detection module of CPDH@cAS (**Figure R11b**).

Figure R11. a) Schematic diagram of detecting tumor recurrence *in vivo*. b) Fluorescence imaging of different tumor cells injected after *in vivo* encapsulation of CPDH@cAS. c) Relative fluorescence quantification of corresponding fluorescence images in b). d) CPDH@cAS detected the signal intensity of different densities of tumor cells at 1 h.

The experimental results showed that the ATP sensor exhibited high sensitivity in detecting tumor cells, as indicated by the increasing fluorescence signal with higher cell counts. This suggests that the ATP sensor has the potential to detect even small amounts of tumor cells (500 cells). Additionally, the ATP sensor was able to respond rapidly and reached its highest value at 1 h. The rapid response of the sensor to tumor cells is an important feature that could potentially enable monitoring the progression of cancer and guide treatment decisions. We believe that this approach effectively characterizes the sensitivity of our DNA hydrogels for detecting tumor cells *in vivo* and provides a more comprehensive evaluation of the DNA hydrogel's performance.

We revised as followed:

“In addition, we characterized the sensitivity of CPDH@cAS for *in vivo* tumor cell detection by injecting different numbers of tumor cells at the fully resected surgical site (Figure S12). The experimental results showed that the ATP sensor generated more fluorescence as the cell count increased, indicating its high sensitivity in detecting tumor recurrence. Moreover, once the tumor cells relapsed, the ATP sensor was able to respond rapidly and reached its highest value at 1 h, indicating its timely nature.” was added to the Results and Discussion section as marked in blue.

“*Sensitivity analysis of early warning signals for detecting tumor recurrence in vivo*: Before using, cells were digested with trypsin and counted on a counting plate. The B16F10 cell suspension was diluted into different amounts (0, 500, 1000, 5000, 10,000 and 20,000 cells) into new tubes. General anesthesia was performed by injecting sodium pentobarbital into the peritoneal cavity of mice, and CPDH@cAS hydrogels were surgically embedded in the axillae of mice. Different amounts of B16F10 cell suspension (50 μ L) were injected into the postoperative site where the CPDH@cAS hydrogel had been embedded. Fluorescence imaging was performed to observe the recovery of fluorescence signal from the detection module of CPDH@cAS. Early warning signals were analyzed by semi-quantitative fluorescence intensity of the ATP sensor.” was added to the Methods section as marked in blue.

Question 3: *This result have some of the limitations of PDT – namely that they can slow tumor growth, but they cannot eliminate tumors. Fig 5.*

Response: Thank you for your valuable comments. It has been reported that PDT alone has a limited inhibitory effect on highly malignant B16F10 tumors. [ACS Nano 13, 11249-11262 (2019)] [Angew Chem Int Ed Engl 59, 1897-1905 (2020)] However, in our postoperative embedded system, PDT was employed not only to kill captured tumor cells but also to partially self-disassemble the hydrogel, promoting the release of DNA immunomodulators. This

synergistic approach between PDT and immunotherapy effectively suppresses tumor recurrence. In our future work, we plan to explore other therapeutic and detection modules as well as post-operative tumor models based on this integrated DNA hydrogel system in greater depth.

Question 4: *The text states “PDL1 aptamer and CpG, together with TAA to induce immunomodulatory factors secretion”. What is the mechanism by which PDL1 would like to immune modulatory factor secretion. It is not clear that these studies really show this direct connection, more that the PDL1 gels did better at limiting tumor growth, but it is not clear that PDL1 directly contributed to the immune modulatory factors.*

Response: Thank you for your precious comments and we definitely agree with you. Our previous discussion of this phrase was imprecise. Recent research has indicated that the combined application of PDL1 aptamer, CpG, and TAA induces a notable increase in the production of pro-inflammatory cytokines. [Adv Mater 31, e1904997 (2019)] [Nano Lett 11, 4509–4518 (2022)] However, the precise mechanism by which PDL1 aptamer modulates immunomodulatory factors remains unclear. In light of this, we have modified the sentence marked in blue to read as follows: " **In addition, reactive oxygen species (ROS) break DNA strands in the hydrogel to release encoded PDL1 aptamer and CpG, which together with TAA promotes sufficient systemic antitumor immunotherapy.**"

Question 5: *The manuscript claims to capture and enrich in situ relapsed tumor cells. No histology of the implant is shown to indicate the presence and enrichment of tumor cells in the implant. The in vitro assay for capture is not well described and I do not understand how it was performed. The assay does not seem to recapitulate the in vivo situation, an invasion assays into the gel would seem to be more appropriate. PDL1 is generally described as a signaling molecule not supporting adhesion, and thus this result is a bit confusing.*

Response 5-1: To better characterize the capture and enrichment of tumor cells in the implant, we performed a new animal experiment to directly image the frozen sections of *in vivo* embedded DNA hydrogel for capture of tumor cells (**Figure R12**). Specifically, B16F10 cells were digested and centrifuged to obtain a cell suspension, which was then incubated with Cell Mask™ Deep Red Plasma at 37°C for 30 min to stain the tumor cells. DNA hydrogels, with or without PDL1 aptamer, were stained with SYBR Green II. The mice were anesthetized, and the CPDH and CDH were embedded in the surgical site where the tumor was previously removed, and then sutured. The tumor cells, labeled with Cell Mask™ Deep Red Plasma, were injected *in situ*, avoiding the direct colocalization with hydrogel. After 2 h, the mice were frozen for 1 h, and

frozen sections were prepared to observe the capture of tumor cells by the encapsulated DNA hydrogel using CLSM.

Figure R12. Frozen section fluorescence imaging of DNA hydrogels (with or without PDL1 aptamer) capturing tumor cells *in vivo*. Green: DNA hydrogel, Blue: cell nucleus, Red: tumor cell membrane. Scale bar: 500 μm.

The experimental results showed that CPDH@cAS (encoding PDL1 aptamers) have strong co-localization with tumor cells compared to CDH@cAS (without PDL1 aptamers), indicating that DNA hydrogel systems encoding PDL1 aptamers can enable efficient capture and enrichment of relapsed tumor cells. Therefore, adding the capture module PDL1 aptamer facilitates the early detection and treatment of the DNA hydrogel system. Thanks to the reviewer again for helping us design this new experiment of an invasion assay to better recapitulate the *in vivo* situation.

We revised as followed:

“To examine the ability of the implanted DNA hydrogel in capturing and enriching tumor cells *in vivo*, we simulated post-operative tumor recurrence by introducing a specific number of tumor cells at the site of the fully excised surgical area (Figure 4b). The experimental results demonstrated that CPDH@cAS (with PDL1 aptamers) exhibited strong co-localization with tumor cells compared to CDH@cAS (without PDL1 aptamers), suggesting that adding the PDL1 aptamer in DNA hydrogel system facilitates the *in vivo* capture and enrichment of tumor cells.” was added to the Results and Discussion section as marked in blue.

“*Analysis of tumor cells captured in vivo:* B16F10 cells were digested and centrifuged to obtain a cell suspension, which was then incubated with Cell Mask™ Deep Red Plasma at 37°C. DNA hydrogels, with or without PDL1 aptamer, were stained with SYBR Green II. The mice

were anesthetized, and the CPDH and CDH were embedded in the surgical site where the tumor was previously removed, and then sutured. The tumor cells, labeled with Cell Mask™ Deep Red Plasma, were injected *in situ*, avoiding the direct colocalization with hydrogel. After 2 h, the mice were frozen for 1 h, and frozen sections were prepared to observe the capture of tumor cells by the encapsulated DNA hydrogel using CLSM.” was added to the Methods section as marked in blue.

Response 5-2: PDL1 is a protein found on the surface of certain types of cancer cells, including B16F10 tumor cells, that can bind to a receptor on immune cells called PD-1, leading to the suppression of the immune response against the cancer. PDL1 aptamers are short, single-stranded DNA molecules that can bind specifically to PDL1 protein. [Chem. Commun. 56, 14653-14656 (2020)] [ACS Nano 16, 18921-18935 (2022)] In this work, PDL1 aptamers are encoded onto the hydrogel matrix to create an aptamer-based capture module. The hydrogel matrix provides a three-dimensional structure for the aptamers to interact with their targets and can be used as a capture and enrichment platform for PDL1-positive tumor cells, such as B16F10 cells, in a complex biological sample. Therefore, the use of PDL1-specific aptamers on a hydrogel matrix can enable the selective capture and enrichment of B16F10 tumor cells for diagnostic or research purposes, and potentially contribute to the development of new cancer therapies.

Response 5-3: We have added a detailed description of the *in vitro* assay process of DNA hydrogel capturing tumor cells in the Results and Discussion section as marked in blue: The CDH and CPDH hydrogels were placed into tubes, followed by the addition of 100 μ L cell culture medium containing variable numbers of tumor cells, then incubated at 37°C for 1 h, washed gently twice with PBS, and subsequently imaged by IVIS Spectrum. The results showed that PDL1 aptamer played a crucial role in the capture and enrichment of tumor cells in DNA hydrogel (Figure 3b, 3c).

Question 6: On page 15, the authors state “In contrast, due to the absence of tumor cell enrichment,”, but they do not show histology that would be necessary to conclude tumor cell enrichment or not.

Response: Thanks for the valuable suggestion. To address this issue, we have added new experiments to examine the ability of DNA hydrogels with or without capture modules to capture cells *in vivo* (Figure R12). The experimental results showed that CPDH@cAS (with PDL1 aptamers) demonstrated higher co-localization with tumor cells compared to CDH@cAS (without PDL1 aptamers). This indicates that DNA hydrogel systems encoding PDL1 aptamers

can effectively capture and enrich released tumor cells. In addition, we think another result in the manuscript can also support this hypothesis. Specifically, we performed experiment to analysis the difference of hydrogel with or with PDL1 aptamer for detecting ATP secreted by tumor cells (**Figure R13**). Three groups of mice with B16F10 tumors on their right armpit were embedded with CPDH@crAS, CDH@cAS, and CPDH@cAS after surgery, respectively. Whole-body fluorescence imaging was subsequently performed at the indicated time points to analyze the results.

Figure R13. CPDH@cAS for tumor recurrence detection. a) Fluorescence images of mice with tumor recurrence after embedding different hydrogel sensors. Tumors are indicated by red circles. b) Relative fluorescence quantification of mice tumors at different time points. *P < 0.05, ****P < 0.0001.

The experimental results showed that the CPDH@cAS group in mice with recurrent tumors showed a strong fluorescence signal *in situ* and rapidly reach maximum at approximately 1 h after implantation. In contrast, due to the absence of tumor cell enrichment, fluorescence in the CDH@cAS group (without capture modules) was slower to reach a plateau and remained at a lower level than in the CPDH@cAS group (with capture modules). We think this result indicate that incorporating the capture module (PDL1 aptamer) in DNA hydrogel system enhances the early detection and treatment of tumor.

Question 7: The authors cite an inhibition rate of 88.1%, but it is not clear what that demonstrates. How is this calculated?

Response: We have added a detailed description of how we calculated tumor inhibition rate of DNA hydrogels in the Results and Discussion section as marked in blue: The tumor biofluorescence intensity and tumor volume of this group of mice were the lowest on day 17, calculated from the tumor biofluorescence intensity of the pairs of mice. Calculation formula of tumor inhibition rate: $TIR = (1 - BI_{T1} / BI_{T0}) \times 100\%$ (TIR: tumor inhibition rate, BI_{T1} : tumor bioluminescence intensity in the treatment group, BI_{T0} : tumor bioluminescence intensity in the control group.) CPDH-Ce6+ $h\nu$ inhibited approximately 88.1% and 56.3% of primary and distal tumors respectively (**Figure R14**). In addition, we calculated the inhibition rates of primary and distal tumors in different groups of mice by the formula in **Table R1**.

Figure R14. a) The primary tumor bioluminescence intensities curves of mice after various treatments (n = 5). b) Comparison of bioluminescence intensity of tumors in different treatment groups on 17th. *P < 0.05, **P < 0.01, ***P < 0.001.

Table R1. Primary and distal tumor inhibition rates (TIR) were calculated for different groups of mice

Primary tumor	Average bioluminescence intensity	TIR
Untreated	14.67×10^5 (BI_{T0})	0%
CPDH-Ce6	11.16×10^5 (BI_{T1})	23.9%
DH-Ce6+hv	5.184×10^5 (BI_{T2})	64.7%
CPDH-Ce6+hv	1.7416×10^5 (BI_{T3})	88.1%
Distant tumor	Average bioluminescence intensity	TIR
Untreated	13.34×10^5 (BI_{T0})	0%
CPDH-Ce6	12.174×10^5 (BI_{T1})	8.7%
DH-Ce6+hv	7.246×10^5 (BI_{T2})	45.7%
CPDH-Ce6+hv	5.83×10^5 (BI_{T3})	56.3%

Question 8: *The paper has many acronyms and it is hard to keep track of them. ICD, and RCA were not defined, and others were defined late in the text.*

Response: We are very grateful for your careful review. We have ensured that where abbreviations first appear in the manuscript they are noted. For example, the note for ICD is immunogenic cell death, and the note for RCA is rolling circle amplification, etc.

Question 9: *The text states “Local presentation of DC cells to tumor antigens ...”, which was unclear. I assume DC cells are just dendritic cells. But the wording of presenting cells to tumor antigens is imprecise.*

Response: Thanks for the valuable suggestion. We mentioned in the manuscript: Local presentation of DC cells to tumor antigens could contribute to the increase of CD8⁺ T cells in distant tumors, which triggers systemic anti-tumor immunity. We have revised this ambiguous sentence to: The local presentation of tumor antigens by DC cells could contribute to the increase of CD8⁺ T cells in distant tumors, which triggers systemic anti-tumor immunity.

Question 10: *The text states “the proportion of Tregs (Foxp3⁺ T cells) was decreased, indicating an enhanced antigen-specific T cell response”. I am not sure how you can say it was antigen specific without sequencing the TCR.*

Response: Thank you for your careful review and we definitely agree with you. We have revised this inappropriate sentence in the manuscript as: The proportion of Tregs (Foxp3⁺ T cells) decreased, indicating that CPDH-Ce6+hv effectively reduced tumor-associated immunosuppression.

Question 11: *The last sentence states “Collectively, these data suggest that hydrogels based on PDT combined with the photoresponsive release of DNA-based immunomodulators can induce a robust immune response and thus trigger the most effective tumor suppression.” Do the authors intend to say that the robust response triggers tumor suppression, it sounds like they are indicating a suppressive immune response at the tumor.*

Response: We have modified this obscure sentence to: In conclusion, these data suggest that photodynamic immunomodulatory DNA hydrogels based on tumor cell capture and detection can provide timely warning of recurrence and trigger an effective immune response, resulting in postoperative tumor suppression.

Question 12: *Authors write ... “suggesting that PDL1 aptamers and CpG play an essential role in the treatment.” The absence of PDL1 leads to growth suppression of 50-70%, so it is not clear that PDL1 is essential.*

Response: Inhibition of the interaction between PD-1 and PDL1 by PDL1 aptamer can enhance the immune response to cancer cells. In addition, in the environment where antigen-presenting cells coexist with tumor cells and T cells, the lack of a co-stimulating tumor microenvironment may also be the reason for its limited efficacy. Therefore, the combination of PDL1 aptamer and CpG can synergistically promote the sustained and specific anti-tumor T cell response in the melanoma model. [Adv Mater 28, 8912-8920 (2016)] [Angew Chem Int Ed Engl 59, 1108-1112 (2020)] [Proc Natl Acad Sci U S A 113, E7240-E7249 (2016)] On the other hand, PDL1 aptamers are essential as they can bind to PDL1 proteins highly expressed on the surface of B16F10, enriching and capturing tumor cells and facilitating the sensitivity of detection of early tumor recurrence.

When investigated the antitumor effect of locally encapsulated CPDH-Ce6 on B16F10 bilateral tumor mice, we took PDL1 aptamer and CpG as a whole to investigate the synergistic effect of DNA immunomodulators on tumor recurrence. The results showed that compared with the hydrogel group DH-Ce6 only with PDT, the inhibition rate of tumor recurrence in the CPDH-Ce6 group was increased by about 23.4%. And the results of the two groups were significantly different, indicating that PDL1 aptamer and CpG played an important role in the treatment.

Question 13: *The manuscript indicates fluorescence with crAS, yet crAS nucleotides had been mutated. This reviewer is not understanding something.*

Response: Thank you for the valuable comment. We have added a description of the phenomenon that mutated crAS still showed fluorescence in the revised manuscript: Since the hydrogel was labelled with Cy5, even quenched with BHQ2, there would still be some background signal. [Adv Mater, 2019, 31(33): e1901885]. [Angew Chem Int Ed Engl, 2021, 61(4): e202111836.] Imaging analysis was performed to reduce the impact on data readout by reducing and stabilizing the baseline of the monitoring signal.

Question 14: *The discussion section has no references. Are there no other early detection systems, aptamer systems, PDT approaches for detection or clearance, PDL1 signaling mechanisms, ... that are worthy of mentioning to provide context for the work that was performed in the manuscript?*

Response: Thanks for the constructive suggestion. We have added the background and outlook for the study in the Discussion section: The monitoring and treatment of postoperative malignancy recurrence is a critical and urgent challenge. Despite the availability of advanced detection and treatment systems, the lack of early warning and treatment specificity remains a significant issue. [Acta Pharm Sin B 11, 1978-1992 (2021)] [Small, e2207154 (2023)] [Nat Med

28, 666-677 (2022)] DNA-based materials have attracted significant interest in the detection and treatment of tumors due to their programmable properties and high biocompatibility. For instance, Li et al. developed a DNA nanodevice capable of spatially selective imaging of dual targets (MMP2/9 and ATP) in the tumor cell microenvironment for detecting tumors. [Angew Chem Int Ed Engl, 2021, 61(4): e202111836.] Tan et al. demonstrated enhanced effectiveness of postoperative immune checkpoint blockade therapy by utilizing nucleic acid aptamers to block PD1/PDL1 interaction. [J. Am. Chem. Soc. 143, 8391–8401 (2021)] Du et al. developed a spherical nucleic acid modified with PDL1 aptamers that, when combined with photodynamic therapy, provided effective photodynamic immunotherapy for the suppression of tumors. [Angew Chem Int Ed Engl, e202214750 (2022)]

Reviewer #3

Comments:

In this study, the authors describe and characterize a novel DNA-based hydrogel that captures tumor cells, identifies ATP signals produced by aggregated tumor cells, and contains PD-L1 aptamers. The stated application of this technology is to be able to detect residual cancer cells, capture them, and improve immunogenicity at the site through photodynamic therapy. This is an interesting approach towards a serious problem of minimal residual disease following surgical resection of a tumor. Overall interesting comment but needs additional experiments to back up the claims the authors make and to demonstrate more proof-of-concept for clinical application.

Response:

Thanks for your appreciation for our work with instructive suggestions. Based on your suggestions, we revised the manuscript one by one as follows and marked **in blue**:

Question 1: *what is the mechanism of tumor capture from PD-L1? Especially using the B16F10 model, which is a more immunogenic melanoma cell line.*

Response: Thank you for the valuable comment. PDL1 (Programmed Death-Ligand 1) is a protein that overexpressed on the surface of many cancer types, such as melanoma (40-100%), non-small cell lung cancer (35-95%), glioblastoma (100%), ovarian cancer (33-80%) and colorectal adenocarcinoma (53%) (**Table R2**). [Clin Cancer Res 18, 6580-6587 (2012)] PDL1 plays a critical role in regulating the immune response by interacting with its receptor PD-1 on immune cells such as T cells. PDL1 aptamers are short single-stranded DNA or RNA molecules that specifically bind to the PDL1 protein. As PDL1 is heavily glycosylated, the smaller size of the PDL1 aptamer in comparison to antibodies makes it a more suitable candidate for binding to

PDL1 proteins expressed on tumor cells. By binding DNA aptamer with the cell membrane protein, cells can be captured by the DNA structure. [Nanoscale 14, 8995-9003 (2022)] [Angew Chem Int Ed Engl 59, 4800-4805 (2020)]

B16F10 is a mouse melanoma cell line that has been extensively used as a preclinical model for studying various aspects of cancer biology, including tumor progression, metastasis, and response to therapy. In recent years, B16F10 has also been used as a model for testing the efficacy of anti-PDL1 treatment. Several studies have shown that treatment with anti-PDL1 antibodies can inhibit the growth and metastasis of B16F10 melanoma in mouse models. [Sci Immunol, 2019, 4 (37): eaau6584] [Sci Transl Med, 2018, 10 (429): eaan3682] [Nature Materials, 2023, 1 (10)] These findings suggest that B16F10 has high PDL1 expression and could be a useful tool for studying the mechanisms underlying the response to anti-PDL1 treatment, as well as for evaluating the efficacy of new immunotherapeutic approaches for cancer.

Table R2. Expression of PDL1 in tumors

Cancer type	% PDL1+	References
Melanoma	40–100	PLoS One 6, e17621 (2011). Cancer 116,1757–66 (2010). Nat Med 8,793–800 (2002). PLoS One 6, e17621 (2011).
Non–small cell lung cancer	35–95	Nat Med 8,793–800 (2002). Clin Cancer Res 10,5094–100 (2004). J Immuno 1187,1113–9 (2011).
Nasopharyngeal cancer	68–100	Mod Pathol 23,1393–403 (2010).
Glioblastoma/mixed glioma	100	Cancer Res 63,7462–7 (2003).
Colon adenocarcinoma	53	Nat Med 8,793–800 (2002).
Esophageal cancer	42	Clin Cancer Res 11,2947–53 (2005).
Pancreatic cancer	39	Clin Cancer Res 13,2151–7 (2007). PLoS One 6, e17621 (2011).
RCC	15-24	Clin Cancer Res 13,1749–56 (2007). Cancer Res 66, 3381–5 (2006). J Nucl Med 50,974–81 (2009).
Breast cancer	31–34	Neoplasia 8,190–8 (2006). Hum Pathol 39,1050–8 (2008).
Lymphomas	17–94 ^a	Blood 114,2149–58 (2009). Clin Cancer Res 11,5708–17 (2005).
Leukemias	11–42	Cancer Immunol Immunother 59,1839–49 (2010) Leukemia 23,375–82 (2009).

To further test the PDL1 expression level on B16F10, we conducted additional experiments to investigate the PDL1 expression in various tumor cells (Hela, 3T3, 4T1, B16, and B16F10) by Western blot analysis (**Figure R15**). Specifically, all types of cells were digested to obtain cell suspensions. The cells were lysed on ice for 40 min in PMSF protease inhibitor and lysis solution (1:100 ratio). bicinchoninic acid protein assay kit was used for protein quantification. 30 μ g of each protein sample was loaded into 12% SDS-PAGE and separated by electrophoresis. Transfer to polyvinylidene fluoride (PVDF) membranes blocked with 5% skimmed dry milk for 1 h. PVDF membranes were incubated with primary antibody at 4°C overnight. After washing the PVDF membrane liberally with TBST (tris buffered saline with 0.1% Tween), the membrane was incubated with secondary antibody at 4°C for 1 h. After washing with TBST three times, the bands were observed with the enhanced chemiluminescence (ECL) kit, and the density of the immunoblot was detected with ImageJ.

Figure R15. The expression levels of PDL1 protein in different tumor cells (Hela, 3T3, 4T1, B16 and B16F10) were analyzed by a) western-blot and b) semi-quantification with Image-J software. ***P < 0.001.

The expression levels of the PDL1 protein were analyzed across various tumor cells, revealing relatively high expression in B16F10 cells. Based on the mechanism discussed before, PDL1 aptamer-rich DNA hydrogels should be able to capture and enrich B16F10 cells and block the interaction between PD1 and PDL1 to interrupt immune escape from tumors. Moreover, melanoma is located in superficial layers and is commonly treated clinically with surgery and photodynamic therapy, thus the B16F10 tumor model was chosen for this study.

We revised as followed:

“Nucleic acid aptamers offer several advantages over antibodies, including large-scale synthesis, ease of modification, and low immunogenicity. [Angew Chem Int Ed Engl, e202214750 (2022)] [Science 334, 1716-1719 (2011)] [Nat Rev Drug Discov 16, 181-202

(2017)] Importantly, the programmable nature of nucleic acid aptamers enables the PDL1 aptamer to be encoded in DNA hydrogel systems. This allows the DNA hydrogel to bind specifically to the PDL1 protein expressed on the surface of tumor cells, facilitating *in situ* capture and enrichment of tumor cells. Besides, we added experiments to examine the expression levels of PDL1 protein in different tumor cells by Western-blot, which could be seen to be relatively highly expressed in B16F10 cells (Figure S8). As a result of these features, we selected the PDL1 aptamer as the cell enrichment module for our study.” was added to the Results and Discussion section as marked in blue.

Question 2: Fig. 3E does not seem to have a wide dynamic range of signal between 10-50 cells/ μ L and statistical analysis of 3F would be more convincing. what if the cell density was increased 10 or 100 fold?

Response: Thanks for the valuable suggestion. We conducted additional experiments to investigate the capacity of DNA hydrogels in detecting tumor cells, as the cell density was gradually increased to 100, 500, 1000, and 5000 cells/ μ L (Figure R16). Briefly, B16F10 cells were digested with trypsin and counted on a counting plate. Subsequently, cells were placed into 100 μ L of PBS to make cell suspension with different cell concentration (0, 10, 20, 30, 40, 50, 100, 500, 1000 and 5000 cells/ μ L) were configured. The prepared CPDH@cAS was incubated with different densities of cells in the medium for 1 h at 37°C. The gel was then removed, washed twice with PBS buffer and transferred to a new tube. Fluorescence imaging was performed to observe the recovery of fluorescence signal from the detection module of CPDH@cAS. Finally, Bruker MR SE software was used to analyze the experimental results.

Figure R16. a) the ATP sensor produced fluorescent images of the detected signal and b) the corresponding fluorescence intensity. c) Linearity plotted between 0-50 cells/ μL .

The experimental results showed that the intensity of the fluorescence signal detected by the ATP sensor enhanced with increasing cell count in the concentration range of 0-5000 cells/ μL , indicating its high sensitivity in detecting tumor recurrence. Moreover, a strong linear correlation was observed in the density range of 0-50 cells/ μL . Additionally, the fluorescence signal detected by the ATP sensor reached a plateau at a cell count of 500 cells/ μL , which may be limited by the total fluorescence intensity of the detection module.

We revised as followed:

“The experimental results showed that the intensity of the fluorescence signal detected by the ATP sensor enhanced with increasing cell count in the density range of 0-5000 cells/ μL , indicating its high sensitivity in detecting tumor recurrence (Figure 3d, 3e). Furthermore, the fluorescence signal detected by the ATP sensor reached a plateau at a cell count of 500 cells/ μL and had a linear correlation ($Y = 66.12X + 4432$, $R^2 = 0.9660$) within a certain range of tumor cell concentrations (0-50 cell/ μL) (Figure 3f).” was added to the Results and Discussion section as marked in blue.

“*In vitro detection of tumor cells:* B16F10 cells were digested with trypsin and counted on a counting plate. Subsequently, cells were placed into 100 μL of PBS to make cell suspension with different cell concentrations (0, 10, 20, 30, 40, 50, 100, 500, 1000, and 5000 cells/ μL) were configured. The prepared CPDH@cAS was incubated with different densities of cells in the medium for 1 h at 37°C. The gel was then removed, washed twice with PBS buffer and transferred to a new tube. Fluorescence imaging was performed to observe the recovery of fluorescence signal from the detection module of CPDH@cAS. Finally, Bruker MR SE software was used to analyze the experimental results.” was added to the Methods section as marked in blue.

Question 3: *control experiment needed for Fig. 3G with laser irradiation of tumor cells in general - otherwise hard to know if anti-proliferative effect is solely driven by hydrogels.*

Response: Thank you for the valuable comment. We have added a control experiment with laser irradiation of DNA hydrogel (without the PDT module) treated tumor cells (**Figure R17**). Specifically, to evaluate the ability of CPDH to damage cells, the cells in the logarithmic growth phase were taken to prepare for cell suspension. B16F10 cell (2×10^3) suspension and CPDH

(100 μ L) in 10% RPMI 1640 complete medium were incubated together at 37°C for 1 h. The resulting hydrogel was washed and then transferred to another tube containing RPMI 1640 medium. After treated with 10 min of laser irradiation, CPDH was digested using 0.01 U of DNase I for 30 min and placed in a fresh medium until the cells formed colonies (approximately 2 weeks). It is worth noting that the cell culture medium needs to be replaced every other day. After the experiment, the cells were fixed in anhydrous methanol for 20 min and stained with 0.1% crystal violet for 15 min, then photographed after washing. The colony formation rate was calculated according to the following formula: (number of colonies in the experimental group/number of colonies in the control group) \times 100%.

Figure R17. a) Images of clone formation and b) corresponding clone formation rates of B16F10 cells after CPDH+hv, CPDH-Ce6, and CPDH-Ce6+hv treatment. Data are means \pm SD (n = 3). ****P < 0.0001.

The results indicated that CPDH-Ce6+hv demonstrated a significant inhibition rate of 49.45% on B16F10 tumor cell cloning. In contrast, the group receiving CPDH+hv (without PDT module) and the group receiving CPDH-Ce6 (without laser irradiation) had minimal impact on the cloning rate of tumor cells.

Question 4: *At face value, the ATP detection with their system is impressive in Figure 4. However, the hydrogels were embedded after a partial resection where the MRD would be high. It is hard to imagine a scenario where the the MRD burden would be so high after a partial resection, and therefore the design of this experiment limits its real-world application. Would like to see how sensitive the hydrogel is at lower concentrations of residual cells (i.e. full resection) and whether or not it can detect them.*

Response: Thanks for the constructive suggestion. To better simulate the ATP levels in the early stages of tumor recurrence, we have added experiments to evaluate the timely detection of ATP

sensor by injecting a series of tumor cells (0, 500, 1000, 5000, 10000 and 20000 cells count) at the fully resected surgical site (**Figure R18a**). Specifically, B16F10 cells were digested with trypsin and counted on a counting plate. Different amounts of B16F10 cell (0, 500, 1000, 5000, 10000 and 20000 cells count) suspensions were injected into the postoperative site where CPDH@cAS hydrogel has been embedded. Fluorescence imaging was performed to observe the recovery of fluorescence signal from the detection module of CPDH@cAS (**Figure R18b**).

Figure R18. a) Schematic diagram of detecting tumor recurrence *in vivo*. b) Fluorescence imaging of different tumor cells injected after *in vivo* encapsulation of CPDH@cAS. c) Relative fluorescence quantification of corresponding fluorescence images in b). d) CPDH@cAS detected the signal intensity of different densities of tumor cells at 1 h.

The experimental results showed that the ATP sensor exhibited high sensitivity in detecting tumor cells, as indicated by the increasing fluorescence signal with higher cell counts. This suggests that the ATP sensor has the potential to detect even small amounts of tumor cells (500 cells). Additionally, the ATP sensor was able to respond rapidly and reached its highest value at 1 h. The rapid response of the sensor to tumor cells is an important feature that could potentially enable monitoring the progression of cancer and guide treatment decisions. We believe that this approach effectively characterizes the sensitivity of our DNA hydrogels for detecting tumor cells *in vivo* and provides a more comprehensive evaluation of the DNA hydrogel's performance.

We revised as followed:

“In addition, we characterized the sensitivity of CPDH@cAS for *in vivo* tumor cell detection by injecting different numbers of tumor cells at the fully resected surgical site (Figure S12). The experimental results showed that the ATP sensor generated more fluorescence as the cell count increased, indicating its high sensitivity in detecting tumor recurrence. Moreover, once the tumor cells relapsed, the ATP sensor was able to respond rapidly and reached its highest value at 1 h, indicating its timely nature.” was added to the Results and Discussion section as marked in blue.

“*Sensitivity analysis of early warning signals for detecting tumor recurrence in vivo:* Before using, B16F10 cells were digested with trypsin and counted on a counting plate. The cell suspension was diluted into different amounts (0, 500, 1000, 5000, 10,000 and 20,000 cells) into new tubes. General anesthesia was performed by injecting sodium pentobarbital into the peritoneal cavity of mice, and CPDH@cAS hydrogels were surgically embedded in the axillae of mice. Different amounts of B16F10 cell suspension (50 μ L) were injected into the postoperative site where the CPDH@cAS hydrogel had been embedded. Fluorescence imaging was performed to observe the recovery of fluorescence signal from the detection module of CPDH@cAS. Early warning signals were analyzed by semi-quantitative fluorescence intensity of the ATP sensor.” was added to the Methods section as marked in blue.

Question 5: *The text connection with "And there was no non-specific signal in the major organs (heart, liver, spleen, lung, and kidney) in all three groups, demonstrating that the hydrogel system has no significant toxic side effects in vivo (Figure 4e)." is not reflected in the figure.*

Response: Thank you for your careful review and we definitely agree with you. We have modified the description as: *And there was no non-specific signal in the major organs (heart, liver, spleen, lung, and kidney) in all three groups, demonstrating that the hydrogel system could be stable and won't release its fluorophore to other organs after being implanted for 48 h. The conclusion that the DNA hydrogel has low toxic effects can be proven by H&E and routine blood test in the major organs of the mice after treatment.*

Question 6: *The metastasis model used in Fig. 5 is not a natural seeding model, nor is it via accepted routes of metastasis for melanoma. It would be better to do this experiment in a spontaneous model of metastasis or other more accurate models of melanoma metastasis.*

Response: Thanks for the constructive suggestion. We have performed new experiments to examine the inhibitory effect of photodynamic immunotherapy of DNA hydrogel systems on tumor metastasis by constructing a standard melanoma metastasis model (**Figure R19a**). [*Adv Mater* 28, 8912-8920 (2016)] [*Adv Mater* 34, e2205950 (2022)] Briefly, B16F10 cells (1×10^7) were injected subcutaneously in the right axilla. Mice were randomly divided into 6 groups and post-operative excision and encapsulation of hydrogel preparations were performed on day 7. B16F10 cells were injected in the tail vein on day 10 and the primary tumor was irradiated daily with a 660 nm laser (0.2 Wcm^{-2} , 10 min). Mice were dissected after day 17 and lungs were removed. Lung metastases were assessed by photo graphs and histological analysis of H&E staining (**Figure R19b**, **Figure R19c**).

Figure R19. Postoperative metastasis inhibition in B16F10 tumor-bearing C57BL/6 mice by laser-irradiated CPDH-Ce6. a) Schematic diagram of the experimental design for the construction and treatment of lung metastasis animal models. b) Representative photographs of lung tissues from mice in different groups. c) Representative H&E of lung sections from mice in different treatment groups. Scale bar: 200 μ m. The dotted circles mark the damaged regions.

The experimental results showed that there were no obvious visible metastatic nodules in CPDH-Ce6+hv-treated mice compared to control and other groups of mice. Based on the experimental results, it appears that the treatment with CPDH-Ce6+hv had a positive effect on preventing postoperative lung metastases in mice. The absence of visible metastatic nodules in the treated mice, as compared to the control and blank hydrogel mice, suggests that the treatment may have been successful in inhibiting the spread of cancer cells.

We revised as followed:

“2.6 Evaluation of DNA hydrogel for suppression of lung metastases after surgical resection of primary tumors

Lung metastasis is another major clinical problem in malignant cancer patients after surgery. To evaluate the effect of DNA hydrogel treatment on lung metastases after surgical resection of primary tumors, we constructed and treated a standard B16F10 lung metastasis model (Figure 7a). Then, lung tissue was collected from each group of mice for analysis and comparison (n = 3). Lung metastases were assessed by photo graphs and histological analysis of H&E staining.

There were no obvious visible metastatic nodules in CPDH-Ce6+hv-treated mice compared to control and other groups of mice, suggesting effective prevention of postoperative lung metastases (Figure 7b, Figure 7c).” was added to the Results and Discussion section as marked in blue.

“*Lung metastasis analysis:* A lung metastasis model was established to evaluate the anti-metastatic effect of CPDH-Ce6. Briefly, B16F10 cells (1×10^7) were injected subcutaneously in the right axilla. Mice were randomly divided into 6 groups and post-operative excision and encapsulation of hydrogel preparations were performed on day 7. B16F10 cells were injected in the tail vein on day 10 and the surgical site was irradiated daily with a 660 nm laser (0.2 Wcm^{-2} , 10 min). Mice were dissected after day 17 and lungs were removed for visual photography and H&E staining.” was added to the Methods section as marked in blue.

Question 7: *Would be interesting to see if immunomodulatory effect holds for TILs at the metastatic site as well, for Figure 6 experiments.*

Response: Thanks for the valuable suggestion. We have added experiments to analyze the level of TILs, including the extent of DC cell maturation and T cell activation, in a model of melanoma metastasis (**Figure R20**). Briefly, to analyze DC cell maturation and T cell infiltration in lung metastasis sites, immunofluorescent staining (CD3/CD8 and CD80/CD86) was performed on mouse lung tissue at the end of lung metastasis model treatment according to the manufacturer's instructions. To analyze the immune cells in the lymph nodes, lymph nodes adjacent to tumor were collected to study the infiltration of mature DCs and T cells in the tumor-draining lymph nodes. Analysis was performed by flow cytometry. All data were analyzed using Flow-Jo software.

Figure R20. a) Immunofluorescence images showing DC (green: CD80⁺, red: CD86⁺) and CTL (green: CD3⁺ T cells, red: CD8⁺ T cells) infiltration in the lungs of different groups. Scale bar: 100 μm. b) Analysis of fluorescence intensity of CD80⁺ and CD86⁺ DC cells, CD3⁺ and CD8⁺ T cells in lung sections (n = 3). c) Representative flow cytometric analysis of DCs (CD80⁺CD86⁺) and d) CTLs (CD3⁺CD8α⁺) in lymph nodes (tumor ipsilateral), and quantitative results of the different groups. *P < 0.05, **P < 0.01, ***P < 0.001, ****P < 0.0001.

The experimental results showed that the proportion of CD3⁺ T cells, CD8⁺ T cells, CD80⁺ DC cells, and CD86⁺ DC cells was significantly increased in the lung tissue of CPDH-Ce6+hv-treated mice, which was 1.7-fold, 6.9-fold, 2.1-fold and 2.8-fold higher, respectively, than that of mice in the untreated group. Moreover, compared to the untreated and DH-Ce6+hv groups, the proportion of CD80⁺CD86⁺ DC cells in the CPDH-Ce6+hv group increased by 2.9 and 1.3-fold, respectively, indicating that CPDH-Ce6+hv could promote the maturation of DC cells in lymph nodes. The proportion of CD3⁺CD8⁺ T cells in the CPDH-Ce6+hv group increased by 4.5 and 1.5-fold, respectively, indicating that CPDH-Ce6+hv could promote the activation of T cells in lymph nodes (Figure 7f). Moreover, the proportions of CD3⁺CD8⁺ T cells and CD80⁺CD86⁺ DC cells in the CPDH-Ce6+hv group were increased by 2.6 and 2.1-fold, respectively, compared to the unirradiated CPDH-Ce6 group, indicating that photoactivated CPDH-Ce6 could exert photodynamic immunotherapeutic effects. In conclusion, the experimental results showed that locally embedded immunomodulatory hydrogels not only suppress post-operative tumor metastasis but also enhance systemic tumor immunity.

We revised as followed:

“To assess the recruitment of immune cells in the lung, we stained frozen lung sections by immunofluorescence followed by CLSM imaging. As shown in Figure 7d and Figure S18, the proportion of CD3⁺ T cells, CD8⁺ T cells, CD80⁺ DC cells, and CD86⁺ DC cells was significantly increased in the lung tissue of CPDH-Ce6+hv-treated mice, which was 1.7-fold, 6.9-fold, 2.1-fold, and 2.8-fold higher, respectively, compared to the untreated group of mice. These findings suggest that the treatment was effective in stimulating the immune response in the mice's lungs, as indicated by the increased presence of T cells and DC cells.

In addition, we collected axillary and inguinal lymph nodes ipsilateral to the tumor and analyzed immune cell levels to assess the systemic anti-tumor immune response triggered by DNA hydrogel treatment. As shown in Figure 7e, compared to the untreated and DH-Ce6+hv groups, the proportion of CD80⁺CD86⁺ DC cells in the CPDH-Ce6+hv group increased by 2.9 and 1.3-fold, respectively, indicating that CPDH-Ce6+hv could promote the maturation of DC cells in lymph nodes. The proportion of CD3⁺CD8⁺ T cells in the CPDH-Ce6+hv group increased by 4.5 and 1.5-fold, respectively, indicating that CPDH-Ce6+hv could promote the activation of T cells in lymph nodes (Figure 7f). Moreover, the proportions of CD3⁺CD8⁺ T cells and CD80⁺CD86⁺ DC cells in the CPDH-Ce6+hv group were increased by 2.6 and 2.1-fold, respectively, compared to the unirradiated CPDH-Ce6 group, indicating that photoactivated CPDH-Ce6 could exert photodynamic immunotherapeutic effects. Taken together, the results suggested that this DNA hydrogel system can induce robust photodynamic therapy and systemic

immune response to inhibit lung metastasis.” was added to the Results and Discussion section as marked in blue.

“Analysis of immunomodulation in metastasis models: To analyze DC cell maturation and T cell infiltration in lung metastasis sites, immunofluorescent staining (CD3/CD8 and CD80/CD86) was performed on mouse lung tissue at the end of lung metastasis model treatment according to the manufacturer's instructions. To analyze the immune cells in the lymph nodes, lymph nodes adjacent to tumor were collected to study the infiltration of mature DCs and T cells in the tumor-draining lymph nodes. Analysis was performed by flow cytometry. All data were analyzed using Flow-Jo software.” was added to the Methods section as marked in blue.

Question 8: In Figure 6F, what type of statistical test was done and were other groups compared?

Response: We performed the statistical analysis of Student's t-test in Figure 6f. Moreover, we have added a comparative analysis of the untreated group and CPDH-Ce6 group with CPDH-Ce6+hv group (**Figure R21**).

Figure R21. Immunocytokine levels in sera of mice isolated 4 days after different treatments. Results are presented as means \pm standard deviation (SD). (n = 3). *P < 0.05, **P < 0.01, ***P < 0.001 determined by Student's t-test.

Question 9: Make abbreviations more clear throughout paper, (e.g. CDH is without the PD-L1 aptamer, not clearly explained in text).

Response: We have ensured that where abbreviations first appear in the manuscript they are noted. For example, the note for CDH is a hydrogel system without PDL1 aptamer, etc.

Question 10: There are some grammatical errors in the text which could use more proofreading.

Response: Thank you for your valuable suggestions. We have critically checked the manuscript for grammatical issues and corrected the inappropriateness. For example, replace the sentence:

we finally chosen 1:20 ratio for the following experiments, in line 22 on page 8 with: For subsequent experiments, we selected the 1:20 ratio of Ce6-cDNA to CPDH to synthesize the integrated DNA hydrogel.

Question 11: *Please soften the language around treatment effect. For example, the hydrogel does not inhibit tumor recurrence and metastasis, but only abrogates it compared to untreated controls.*

Response: We have moderated the therapeutic effects around the language in the revised manuscript. For example, after confirming the inhibitory effects of CPDH-Ce6 on tumor recurrence and metastasis in comparison to the untreated group, we further investigated the specific immune response elicited at the tumor site., etc.

Question 12: *The description of CD4+CD8+ T cells make it seem like the authors are looking for co-localized cells, but actually describing separate T cells.*

Response: We are very grateful for your careful review. We modified the CD4⁺CD8⁺ T cells in the manuscript to CD4⁺ T cells and CD8⁺ T cells. And we have double-checked all the spelling issues in the whole manuscript to make sure there are no other mistakes.

Question 13: *Language around claims throughout the text needs to be softened, such as "most effective tumor suppression", most effective compared to what?*

Response: Thanks for the reminding. We have carefully revised the manuscript for inappropriate language expressions and have again weighed the use of sentences: In conclusion, these data suggest that photodynamic immunomodulatory DNA hydrogels based on tumor cell capture and detection can provide timely warning of recurrence and trigger an effective immune response, resulting in postoperative tumor suppression., etc.

REVIEWER COMMENTS

Reviewer #1 (Remarks to the Author):

The authors have addressed most of my concerns.

Reviewer #2 (Remarks to the Author):

The authors have performed a number of experiments to supplement the initial manuscript. The system has many features, though there remains concerns that the text overstates the results. I am overall enthusiastic about the results, yet the claims need to be appropriately modified and a couple of additional pieces of data would strengthen the manuscript (quantification of metastasis). Specific comments are:

1. The model is not truly a model of recurrence. The cells are injected into the site of a primary tumor resection. The 6 day lifetime of the probe is limiting for this, but the study could be performed in a different way.
2. The in vivo studies that refer to capture of tumor cells, is not really representing captures in the sense of what in vivo capture normally represents, i.e., circulating cells are captured at the implant. The implant is capturing cells that are injected into the site.
3. Fig R12 is not very compelling as presented. The resolution is too low to see individual cells.
4. Fig R7. The CPDH-Ce6+hv condition indicates a circle that suggests a site of metastasis. Yet the text states there is not obvious visible metastatic nodules. There are only visual images shown and not quantification.
5. The final sentence of discussion states "great potential for clinical translation". I think this is an overstatement at this point. Again the data is interesting, but too early to state great potential.
6. The discussion remains relatively sparse. They added a couple of sentences in the initial paragraph, and then the remainder of the discussion is largely a restatement of the results.
7. A potential overstatement of mechanism: "... due to the absence of PDT and the inability to promote the release of PDL1 aptamer and CpG." I don't believe you have done the measurements to directly link PDL1 release and tumor suppression. I think they are correlated, but not mechanistic.
8. Line 378 - uses biofluorescence and the couple lines later uses luminescence. I had thought this was luminescence for the B16F10 and was confused.

Reviewer #3 (Remarks to the Author):

Comprehensive response to concerns. Recommend accept without new revisions.

Response to reviewers' comments

Reviewer #1

Comments:

The authors have addressed most of my concerns.

Response:

We really appreciate the valuable comments from reviewers which helped us a lot in improving the quality of our manuscript.

Reviewer #2

Comments:

The authors have performed a number of experiments to supplement the initial manuscript. The system has many features, though there remains concerns that the text overstates the results. I am overall enthusiastic about the results, yet the claims need to be appropriately modified and a couple of additional pieces of data would strengthen the manuscript (quantification of metastasis).

Response:

Thanks for pointing out the remaining concerns. We have thoroughly revised the manuscript to avoid the overstate of the results and added some data (quantification of metastasis) to further strengthen the manuscript. Thank you again for the valuable comments.

Question 1: *The model is not truly a model of recurrence. The cells are injected into the site of a primary tumor resection. The 6 day lifetime of the probe is limiting for this, but the study could be performed in a different way.*

Response: Thanks for pointing out the concern. The study was performed in a different way to more accurately control the number of tumor cells presented in the primary tumor resection niche. According to our knowledge, the most widely applied model of tumor recurrence is the partial resection of tumor, the experiments performed in Figure 5 was following the standard protocol. [Nat. Commun. 13, 4553 (2022)] [Nat. Nanotech. 14, 89–97 (2019)] [Nat. Biomed. Eng. 1, 0011 (2017)] However, in tumor resection

surgery, it's challenging to accurately control the number of residual tumor cells, which is the reason why we designed the alternative approach. The stability of DNA hydrogel and ATP probes are also major concerns and need to be improved. In our future studies, we will explore more detection probes or methods to enhance the lifetime of probes for detecting tumor recurrence.

The manuscript was revised as followed:

“To examine the ability of implanted DNA hydrogel for retaining small number of tumor cells within the local environment, we constructed a postoperative model by introducing a specific number of tumor cells into the completely resected surgical area (Figure 4b and Figure S11).” was added to the Results and Discussion section (lines 307~310) as marked in blue.

Question 2: *The in vivo studies that refer to capture of tumor cells, is not really representing captures in the sense of what in vivo capture normally represents, i.e., circulating cells are captured at the implant. The implant is capturing cells that are injected into the site.*

Response: We agree with the reviewer's comments that the “capture of tumor cells” typically implies “the circulating cells are captured at the implant”. In our study, the process is more like the recurrent tumor cells being retained at the post-surgery site, which was mimic by a partial resection model in Figure 5. To further study the ability of designed DNA hydrogel for immobilizing and sensing small number of tumor cells, known number of tumor cells were injected into the post-surgery tumor site, as a supplementary experiment.

Question 3: *Fig R12 is not very compelling as presented. The resolution is too low to see individual cells.*

Response: We appreciate your valuable comment on Figure R12. Since it's a frozen tissue section of implant DNA hydrogel, the figure resolution was limited. We tried to obtain higher-resolution images of Figure R12 to see individual cells and the enlarged figure was shown in Figure R-R1.

Figure R-R1. Frozen section fluorescence imaging of DNA hydrogels (with or without PDL1 aptamer) capturing tumor cells *in vivo* and local magnification images. Green: DNA hydrogel, Blue: cell nucleus, Red: tumor cell membrane. Scale bar: 500 μm (top) and 100 μm (bottom).

Compared to CDH@cAS (without PDL1 aptamer), CPDH@cAS (with PDL1 aptamer) showed obvious co-localization with tumor cells, indicating that the addition of PDL1 aptamer to the DNA hydrogel system facilitates the *in vivo* capture and enrichment of tumor cells.

Question 4: Fig R7. The CPDH-Ce6+hv condition indicates a circle that suggests a site of metastasis. Yet the text states there is not obvious visible metastatic nodules. There are only visual images shown and not quantification.

Response: Thanks for the suggestions regarding Figure R7 in our manuscript. We have revised the text in the manuscript as “CPDH-Ce6+hv could better suppress tumor metastatic in the lung compared to other experimental groups” to avoid the

overstatement of the results.

Besides, we have performed quantification of tumor metastasis in the revised manuscript following standard protocol [Nature 606, 992-998 (2022)] [Advanced Functional Materials 32, (2022)] [Adv Mater 30, e1706719 (2018)] to provide a more comprehensive analysis (Figure R-R2a). Specifically, the fixed lung tissue was examined meticulously under a high-resolution stereomicroscope to identify and metastatic nodules (Figure R-R2b, R-R2c). Moreover, fixed tissue was embedded in paraffin and sections were prepared, which were subsequently subjected to a histological staining technique with hematoxylin and eosin. The metastases present in each section were carefully counted by using a precise light microscope. In addition, the surface area occupied by the metastatic lesions was quantified using Image-J software (Figure R-R2d, R-R2e).

Figure R-R2. a) Schematic diagram of the experimental design for the construction and treatment of lung metastasis animal models. b) Representative photographs of lung tissues from mice in different groups. c) The calculated lung metastasis nodules of mice after various treatments. d) Representative H&E of lung sections from mice in different treatment groups. Scale bar: 200 μm . The dotted circles mark the damaged regions. e) Quantification of the percentage of lung area occupied by metastases in H&E-stained sections (n = 3). Data are means \pm SD. **P < 0.01, ***P < 0.001 determined by Student's t-test.

The experimental results showed that CPDH-Ce6+*h* ν treatment greatly reduced the number of lung metastases. Histological analysis of lung sections further showed that CPDH-Ce6+*h* ν resulted in a significant reduction in the number and size of metastases compared to the untreated group.

The manuscript was revised as follows:

“CPDH-Ce6+*h* ν -treated mice showed much less metastatic nodules in the lungs compared to control and other groups, suggesting that the DNA hydrogel could effectively prevent postoperative lung metastases (Figure 7b, Figure 7c). H&E staining results showed a consistent tendency (Figure 7d, Figure S19).” was added to the Results and Discussion section (lines 472~476) as marked in blue.

“The fixed lung tissue was examined meticulously under a high-resolution stereomicroscope to identify and metastatic nodules. Moreover, fixed tissue was embedded in paraffin and sections were prepared, which were subsequently subjected to a histological staining technique with hematoxylin and eosin. The metastases present in each section were carefully counted by using a precise light microscope. In addition, the surface area occupied by the metastatic lesions was precisely quantified using Image-J software.” was added to the Methods section (lines 820~826) as marked in blue.

Question 5: *The final sentence of discussion states "great potential for clinical translation". I think this is an overstatement at this point. Again the data is interesting, but too early to state great potential.*

Response: Thank you for the suggestion. We agree with the reviewer that it's too early to discuss the clinical translation potential of the DNA hydrogel and have revised the final sentence as "In summary, this post-surgical *in situ* embedded DNA hydrogel has shown promising results in early warning of post-operative tumor recurrence and timely treatment, which could be further expended with different capture, detection and therapy modules" to avoid the overstatement (lines 556~559).

Question 6: *The discussion remains relatively sparse. They added a couple of sentences in the initial paragraph, and then the remainder of the discussion is largely a restatement of the results.*

Response: Thank you for encouraging us to discuss more deeply about our results. We have revised the Discussion section to try to provide more insightful thoughts as (lines 531~559):

"The surveillance and management of postoperative malignancy recurrence represent a critical and pressing challenge within the realm of oncology. Despite notable advancements in detection and treatment modalities, the absence of early warning signs and treatment specificity remains a significant concern.[Acta Pharm Sin B 11, 1978-1992 (2021)] [Small, e2207154 (2023)] [Nat. Med. 28, 666-677 (2022)] Effective identification of tumor recurrence at an early stage is hindered by the limited number of recurrent tumor cells and low concentrations of biomarkers.[ACS Nano, 3c01404 (2023)] [Advanced Functional Materials, 2300199] Moreover, the inadequate immunogenicity and immunosuppression at the site of tumor resection contributes to tumor progression, often resulting in missed treatment opportunities as tumors advance to intermediate or advanced stages.[Journal of Controlled Release 285, 56-66 (2018)] [Small 16, 2004905 (2020)] Consequently, there is an urgent need to develop more sensitive techniques for detecting tumor recurrence and preventing secondary metastasis subsequent to primary tumor resection.

In this work, we developed a post-operative embedded DNA hydrogel for local enrichment of early relapsed tumor cells to achieve timely photodynamic immunotherapy treatment. We found that as the PDL1 aptamer binding site on the hydrogel increased, its capture rate of tumor cells rose from 16.3% to 65%. Moreover, the fluorescence signal of the modified ATP sensor was stronger as the number of cells captured by the hydrogel increased, and there was a linear relationship between the two at a range of tumor cell concentrations (0-50 cells/ μ L). This provides a new perspective to enhance the detection and treatment of postoperative tumor recurrence by effectively capturing or recruiting recurrent tumor cells.

In addition, this photo-responsive self-disassembled immunomodulatory DNA hydrogel triggered immunotherapy at the surgical site and elicited a systemic immune response in a mice melanoma tumor model, with an inhibition rate of approximately 88.1% against recurrent tumors. These results highlight the potential of a spatiotemporally controlled photodynamic immunotherapy as a promising approach to enhance postoperative tumor treatment. In summary, this post-surgical *in situ* embedded DNA hydrogel has shown promising results in early warning of post-operative tumor recurrence and timely treatment, which could be further expended with different capture, detection and therapy modules.”

Question 7: *A potential overstatement of mechanism: " ... due to the absence of PDT and the inability to promote the release of PDL1 aptamer and CpG." I don't believe you have done the measurements to directly link PDL1 release and tumor suppression. I think they are correlated, but not mechanistic.*

Response: Thank you for the suggestion. We agree that the mechanism between release of PDL1 aptamer and tumor suppression was not clear. The sentence was removed to avoid the overstatement.

Question 8: *Line 378 - uses biofluorescence and the couple lines later uses luminescence. I had thought this was luminescence for the B16F10 and was confused.*

Response: Thank you for pointing out the inconsistency. The term should be bioluminescence. The whole manuscript has been thoroughly checked to avoid similar

mistake.

Reviewer #3

Comments:

Comprehensive response to concerns. Recommend accept without new revisions.

Response:

We really appreciated the comments provided by reviewers which helped us a lot in improving the manuscript quality.